# Drivers and Impacts of the Record-Breaking 2023 Wildfire Season in Canada

Piyush Jain [1] ✉, Quinn E. Barber [1], Stephen W. Taylor[2], Ellen Whitman[1], Dante Castellanos Acuna[3,4], Yan Boulanger[5], Raphaël D. Chavardès[6], Jack Chen [7], Peter Englefield[1], Mike Flannigan[4], Martin P. Girardin[5], Chelene C. Hanes[8], John Little[1], Kimberly Morrison[2], Rob S. Skakun[1], Dan K. Thompson [8], Xianli Wang [1] & Marc-André Parisien [1]

The 2023 wildfire season in Canada was unprecedented in its scale and intensity, spanning from mid-April to late October and across much of the forested regions of Canada. Here, we summarize the main causes and impacts of this exceptional season. The record-breaking total area burned (~15 Mha) can be attributed to several environmental factors that converged early in the season: early snowmelt, multiannual drought conditions in western Canada, and the rapid transition to drought in eastern Canada. Anthropogenic climate change enabled sustained extreme fire weather conditions, as the mean May–October temperature over Canada in 2023 was 2.2 °C warmer than the 1991–2020 average. The impacts were profound with more than 200 communities evacuated, millions exposed to hazardous air quality from smoke, and unmatched demands on fire-fighting resources. The 2023 wildfire season in Canada not only set new records, but highlights the increasing challenges posed by wildfires in Canada.

The 2023 wildfire season in Canada was one of superlatives and broken records. At approximately 15 Mha, the area burned was by far the highest since the start of comprehensive national reporting (c. 1972), shattering the previous record of 6.7 Mha in 1989[1]. Four provinces and territories registered their highest recorded annual area burned, with one fire complex reaching over 1 Mha, larger than any fire since 1950. Over 200 communities had to be evacuated—some twice—under stressful or logistically complex circumstances. There were several fatalities, and hundreds of homes and important infrastructure were destroyed. The smoke produced by Canadian wildfires had widespread consequences, affecting not only nearby communities, but also major population centers >1000 km from the wildfires, notably in southern

Canada and on the east coast of the USA[2]. An unprecedented contingent of wildland firefighters was deployed to respond to the wildfires and international help was obtained from 12 countries and the European Union. The aftermath of the 2023 fire season will likely affect communities and ecosystems for years, if not decades to come.

The 2023 fire season was enabled by record-setting climate. Canada's climate has been warming at twice the global rate: the mean annual temperature has increased 1.7 °C since 1948, with larger increases at high latitudes and during winter and spring[3]. Nationally, the annual area burned in the country has been increasing since the 1950s[4]. While 2023 was unique in both magnitude and character, scientists and managers have long anticipated the growing potential for

[1]Northern Forestry Centre, Canadian Forest Service, Natural Resources Canada, Edmonton, AB T6H 3S5, Canada. [2]Pacific Forestry Centre, Canadian Forest Service, Natural Resources Canada, Victoria, BC V8Z 1M5, Canada. [3]Department of Renewable Resources, University of Alberta, Edmonton, AB T6G 2H1, Canada. [4]Natural Resource Science, Thompson Rivers University, Kamloops, BC V2C 0C8, Canada. [5]Laurentian Forestry Centre, Canadian Forest Service, Natural Resources Canada, 1055 du P.E.P.S., Stn. Sainte-Foy, QC G1V 4C7, Canada. [6]Atlantic Forestry Centre, Canadian Forest Service, Natural Resources Canada, Fredericton, NB E3B 5P7, Canada. [7]Air Quality Research Division, Atmospheric Science and Technology Directorate, Environment and Climate Change Canada, Ottawa, ON K1V 1C7, Canada. [8]Great Lakes Forestry Centre, Canadian Forest Service, Natural Resources Canada, Sault Ste. Marie, ON P6A 2E5, Canada. ✉e-mail: piyush.jain@nrcan-rncan.gc.ca

increased fire activity[5,6]. The length of the wildfire season—the window of opportunity for wildfire to ignite and grow—has been steadily increasing due to warming over the last 50 years[4,7,8]. Projections indicate more frequent and intense fire-conducive weather, especially in the central and western part of the country[9,10]. Moreover, recent wildfire seasons have been associated with extreme weather phenomena affecting an uncommonly large area. For example, the 2021 "heat dome" that covered much of northwestern North America brought record temperatures and enabled one of the worst wildfire seasons of the last century in the province of British Columbia[11], contributing to the abrupt increase in fire activity in the province since ~2000[12]. Undoubtedly, the increasing potential for wildfires in Canada is symptomatic of the changing climate; in fact, it was estimated that the heat dome would have been 150 times less likely to occur without the ongoing anthropogenic climate disruption[13].

Despite this growing body of knowledge about changing fire patterns, the 2023 wildfire season has challenged our understanding of wildland fire in Canada. Canada is about the width of a Rossby wave, the planetary wave pattern[14] associated with meridional flow of the Jet Stream[15]. Severe fire weather is more likely to develop on one side of a wave, leading typical active fire years to occur regionally, rather than throughout the country[16]. Nationwide fire activity, affecting nearly all forested areas in a single season as experienced in 2023 was not anticipated until later in the century based on climate change projections[5]. However, recent studies report an earlier-than-expected emergence of anthropogenic climate change in many parts of North America[17,18]. In fact, a formal climate attribution study demonstrated that anthropogenic climate change strongly contributed to the 2023 wildfire activity in Quebec[19], and that the current levels of activity are enabled by global warming in western Canada[20,21]. Equally surprising in 2023 was the relentless fire activity: except for a handful of days, many large wildfires burned without respite from late April to early October, and several regions underwent two or more significant surges in fire activity. Uncommon fire behavior, including a pyro-tornado in British Columbia (https://www.uwo.ca/ntp/), and a record number of pyro-ocumulonimbus events were observed due to the frequency and intensity of extreme weather and fire intensity. Consequently, large areas of less-flammable vegetation, such as broadleaf-dominated forests and recently burned stands, were affected by wildfires.

Here, we provide an overview of this exceptional year, and examine the main drivers and impacts of the record-breaking 2023 wildfire season in Canada. Specifically, we: (i) describe the spatio-temporal patterns of fire activity, (ii) investigate the drivers of fire activity (i.e., weather conditions, ignition source, and fire duration), and (iii) examine the wildfires' observed and potential impacts on people and ecosystems. To achieve this, we use several datasets quantifying wildfire activity, weather and climate, ignitions, fire management response, and societal impacts. We interpret these observations relative to past wildfire seasons, both nationally and regionally. Finally, we discuss the globally relevant impacts of this extraordinary event in the context of rapid climate change.

## Results and Discussion
### Overview of the wildfire season
The 2023 wildfire season extended across seven months (Fig. 1), following early snowmelt that coincided with record-breaking heat. Many of these wildfires caused evacuations, loss of homes, and infrastructure; several continued to burn for the rest of the fire season, reaching very large sizes (e.g., $\geq 10^5$ ha). The fire season abruptly began in mid-April with an evacuation in southern British Columbia. Shortly afterward, hundreds of mainly human-caused wildfires started in Alberta in early May. In late May, fast-spreading wildfires on the eastern coast—an area where large wildfires are relatively rare—led to the evacuation of thousands of people from several communities in Nova Scotia and the destruction of hundreds of buildings outside of Halifax

and in southern Nova Scotia. In early June, multiple convective cells with associated lightning ignited a string of fires across south-central Quebec, followed by another series of wildfires three weeks later that ignited to the north of the initial ones. The Quebec wildfires burned for weeks and produced a colossal plume of smoke that caused severe smoke pollution in several major cities in Canada and the USA, eventually spreading throughout much of the northern hemisphere[2]. In midsummer, large wildfires burning in the Northwest Territories caused extensive structural losses and the evacuation of about 70% of the population, including the territorial capital city of Yellowknife. In mid-August, wildfires across British Columbia also burned close to communities, destroying hundreds of structures in south-central British Columbia. Embers transported over 3 km across Okanagan Lake started spot fires that threatened the city of Kelowna. Across Canada, there were many rapidly spreading crown fires over the fire season with extreme fire intensity and towering convection columns. Extreme fire behavior and instability generated 140 pyro-cumulonimbus events in the country, 83% of the total observed globally in 2023 (D. A. Peterson, pers. comm.). As a testament to the widespread and sustained nature of the 2023 wildfire activity, September 22nd—usually a quiet time at the end of the season—was the largest single-day area burned in Canada (~440,000 ha) since satellite records began in 1972.

### Record-breaking annual area burned
The 2023 Canadian fire season was remarkable due to coast-to-coast fire activity lasting from mid-April to late October, resulting in a record-breaking total area burned of approximately 15 Mha, corresponding to around 4% of Canada's forest area and more than seven times the historical national average. At the level of individual provinces and territories (Table S1), Quebec, the Northwest Territories, British Columbia, and Alberta all registered a record year for total area burned. In comparison with the 1986–2022 period, the area burned in 2023 was more than double the previous record of 6.7 Mha burned in 1989 (Fig. 2). The widely reported preliminary figure of 18.5 Mha burned is based on active fire data from the fire management agencies for the National Fire Situation Report (CIFFC, https://ciffc.net/). The agencies provide invaluable real-time information during the fire season, but the data are considered to be preliminary estimates for daily operations. Final quality control to account for water bodies and unburned vegetation islands is performed postseason. Our revised estimate is based on a hybrid NBAC-M3 dataset, combining ~12 Mha mapped at high resolution with Landsat and Sentinel-2 satellite imagery and ~3 Mha of perimeters estimated from thermal anomalies.

There were about ~6700 reported wildfire ignitions in 2023, which represents—surprisingly—a lower number than the average of about 8000 wildfires per year[4]. Lightning ignited 59% of the wildfires and lightning-caused wildfires accounted for 93% of the total area burned (1959–2015 average = 91%)[4]. Lightning storms that caused multiple simultaneous ignitions that are beyond initial attack capacity led to several large fires, some of which eventually coalesced. Lightning fires that started on four days, May 13th, May 27th, June 1st and July 5th, were responsible for 30% of the total annual area burned. Interestingly, 2023 had the third lowest overall lightning detections since 1998. The lightning-ignition efficiency was high due to extensive areas of dry, receptive ground fuels coinciding with the specific timing and location of lightning events[22], a phenomenon that is expected to increase as a function of the ongoing warming and drying[23]. Human-caused ignitions were responsible for a comparatively low proportion of the total area burned (7%) in 2023, though it is difficult to confidently assign a human cause to some fires due to ongoing investigations and the eventual intermingling of individual fires with multiple ignition sources in large wildfire complexes. These ignitions were

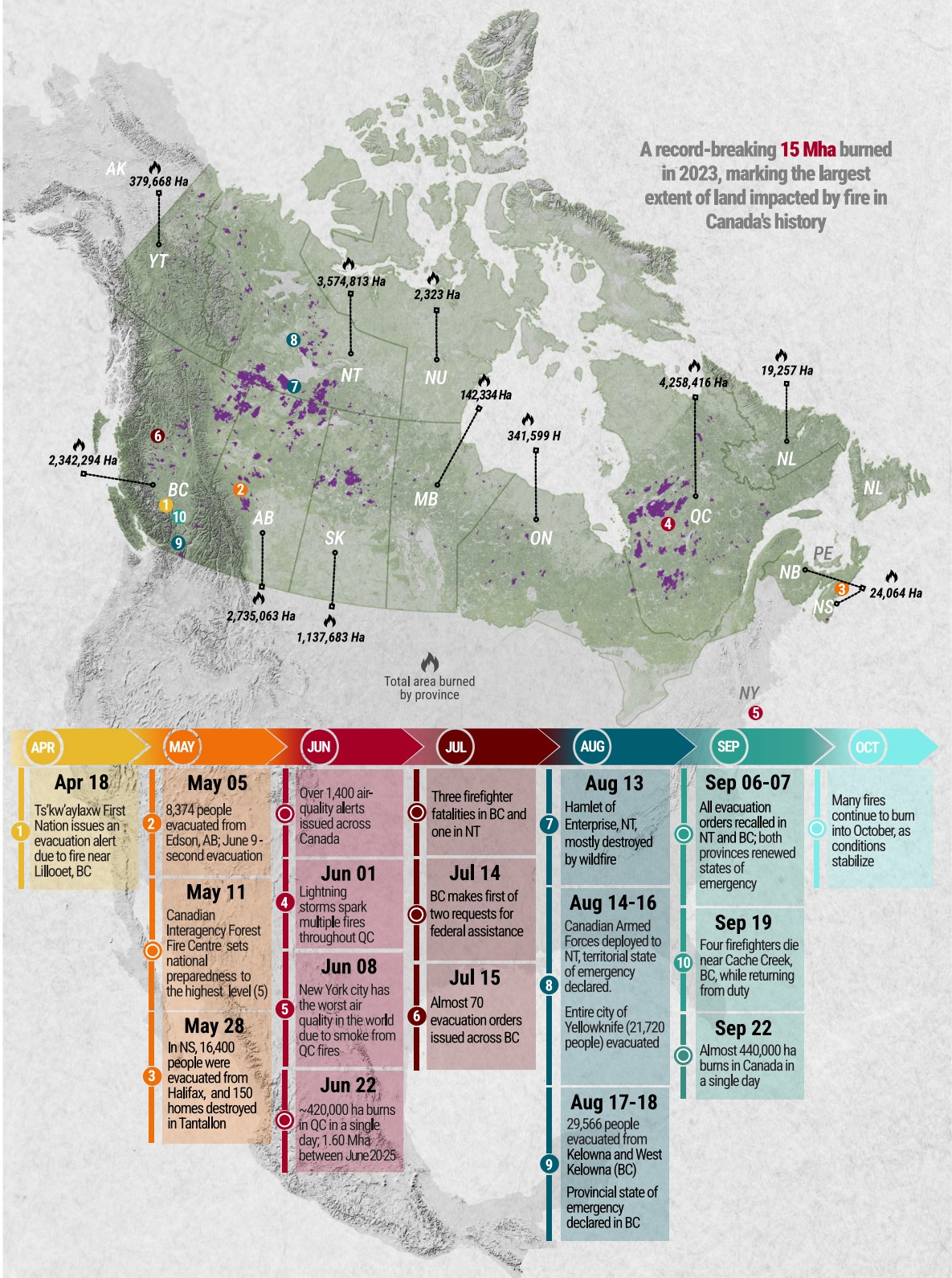

**Fig. 1 | Overview of 2023 fire season.** Infographic showing NBAC-M3 mapped fires, regional area burned and timeline and location of key events that occurred during the Canadian 2023 wildfire season. Background texture designed by Freepik (www.freepik.com).

numerous early in the season before the greening of vegetation and the prevalence of lightning storms. Due to their proximity to communities and infrastructure, human-caused ignitions are associated with a disproportionately large fraction of evacuations and loss of structures[24,25].

**Early season drought conditions**

In 2023, much of Canada faced a significant moisture deficit at the onset of the fire season, primarily attributed to prolonged drought and early snowmelt. The country was snow free considerably earlier than the 2004–2022 average, particularly in northern and west-central

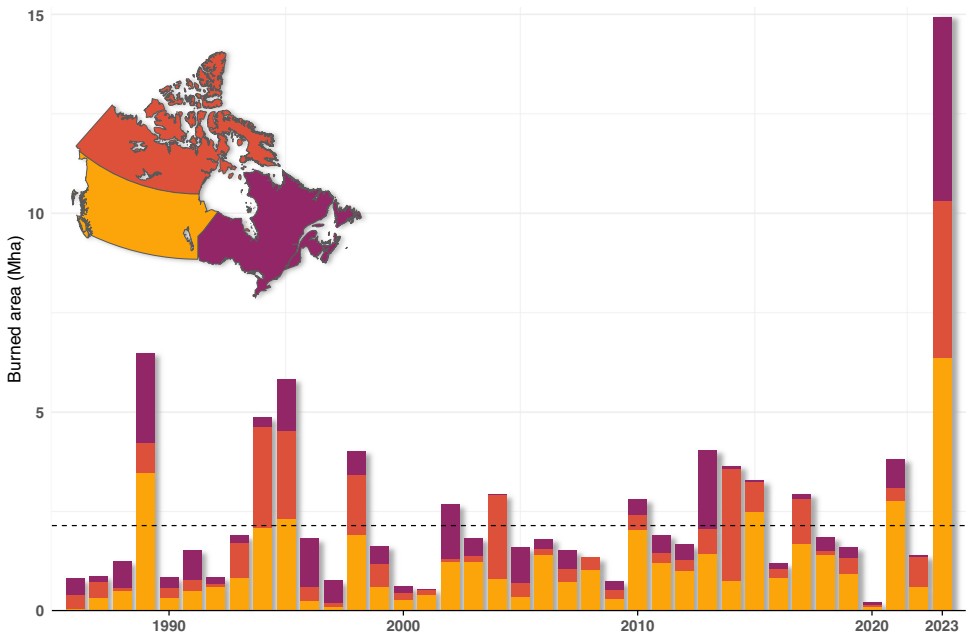

**Fig. 2 | Annual area burned time series for Canada.** Annual area burned for Canada from the National Burned Area Composite (NBAC; 1986–2022) and NBAC-M3 (Natural Resources Canada, 2023b) datasets. During 2023, 15 Mha burned, compared to the annual mean of 2.1 Mha (1986–2022, dashed line). The next largest annual area burned occurred in 1989 with 6.7 Mha.

Canada (British Columbia, Alberta, and Saskatchewan), and along the east coast (Fig. 3 a, b). The maximum root zone soil moisture drying rate increased in western Canada in early May, followed by rapid, intense drying in eastern Ontario and southern Quebec in late May and early June (Fig. 3 c, d). Standardized anomalies of the Canadian Fire Weather Index (FWI) System's Drought Code (DC), show similar trends, with severe drought conditions in some northern and western areas in May, and new-onset drought conditions in Quebec and Nova Scotia in June (Fig. 3d, e). Early-season drought is a common occurrence in western Canada[26], due to persistent drought carried over from the previous year and exacerbated by a low winter snowpack[27] (Fig. S1). In contrast, the 2023 fire season started with near-average levels of soil moisture following snowmelt in the eastern provinces, but above-average temperatures and rapid drying caused what could be described as a 'flash drought', an emerging phenomenon that we are only beginning to understand[28].

**Persistent extreme fire weather**

Although wildfire activity is influenced by many factors, the 2023 season was largely driven by extreme weather enabled by anthropogenic climate change. Canada experienced extreme fire weather conditions in 2023, recognized as the hottest year on record globally[29]. Mean fire season daily temperature, precipitation, and vapor pressure deficit (VPD) anomalies confirm that it was significantly hotter and drier in Canada than average (Fig. 4). The mean May to October fire season temperature was 2.2 °C warmer than the 1991–2020 average, with higher values in northern Canada and average-to-below average values in southern Ontario, Quebec, and New Brunswick. This anomaly is already comparable to what is expected in summertime temperature increases over Canada for ~2050, based on projections from CMIP6 climate models under the SSP5-8.5 scenario (https://climate-scenarios. canada.ca/?page=cmip6-scenarios). VPD is a good predictor of fuel moisture[30] and area burned regionally[31]. Many daily mean temperature and VPD extremes were observed across the country in 2023; for example, on July 7 the temperature at Norman Wells, NWT (65 °N), just south of the Arctic Circle, reached 38 °C. Precipitation deficits were widespread, particularly in regions of western and

northern Canada and western Quebec, all areas that experienced unusually large wildfires during 2023.

Fire growth is largely determined by the coincidence of dry fuels, ignitions, and the occurrence of extreme weather conditions[32,33]. Here, we define extreme fire weather when the Fire Weather Index (FWI) exceeds the climatological 95th percentile of FWI calculated at each location (FWI$_{95}$). The FWI is a meteorological-based measure of overall fire danger used extensively in fire management[34]. In 2023, there was a strong correlation between the forested area where FWI exceeded FWI$_{95}$ and the daily burned area in each province and territory, particularly in western Canada (Fig. 5 and Table 1). Because conditions were extreme for many regions of the country for an extended period of time (e.g., 2–5 months), this enabled the synchronous burning observed in both western and eastern Canada. Synchronous extreme fire weather conditions across large areas has previously been identified as a proxy for constraints on fire suppression resources[35,36]. The highest number of extreme fire weather days occurred in northeastern British Columbia, northern Alberta, and southern Northwest Territories (Fig. 6a). Nationwide, fire weather was most extreme in May and June (Fig. 6b), as a result of significant spring drought conditions over much of the country (Figs. 3 and 4). When examining the mean proportion of daily forested area exceeding FWI$_{95}$, 2023 was also the most extreme fire weather year since at least 1940 (Fig. 6d). Similar relationships were also found for two other FWI System components[34], the Initial Spread Index (ISI) and the Buildup Index (BUI) (Fig. S3). Although drought conditions dominated much of the 2023 fire season, days of extreme fire growth were also related to the presence of high near-surface winds, as evidenced by the largest growth days typically coinciding with short-lived (1–3 days) surges in the FWI (Fig. S6).

Large-scale atmospheric circulation patterns have a strong influence on weather extremes. Atmospheric ridges and the associated positive 500-hPa geopotential height anomalies are associated with fire conducive weather, such as 'blocking' highs[37,38]. Blocking can be characterized by persistent positive anomalies in 500-hPa geopotential heights[39]. Sharma et al.[40] found that persistent positive anomalies in North America were associated with positive fire weather anomalies and a seven-fold increase in the likelihood of wildfire ignitions. In 2023,

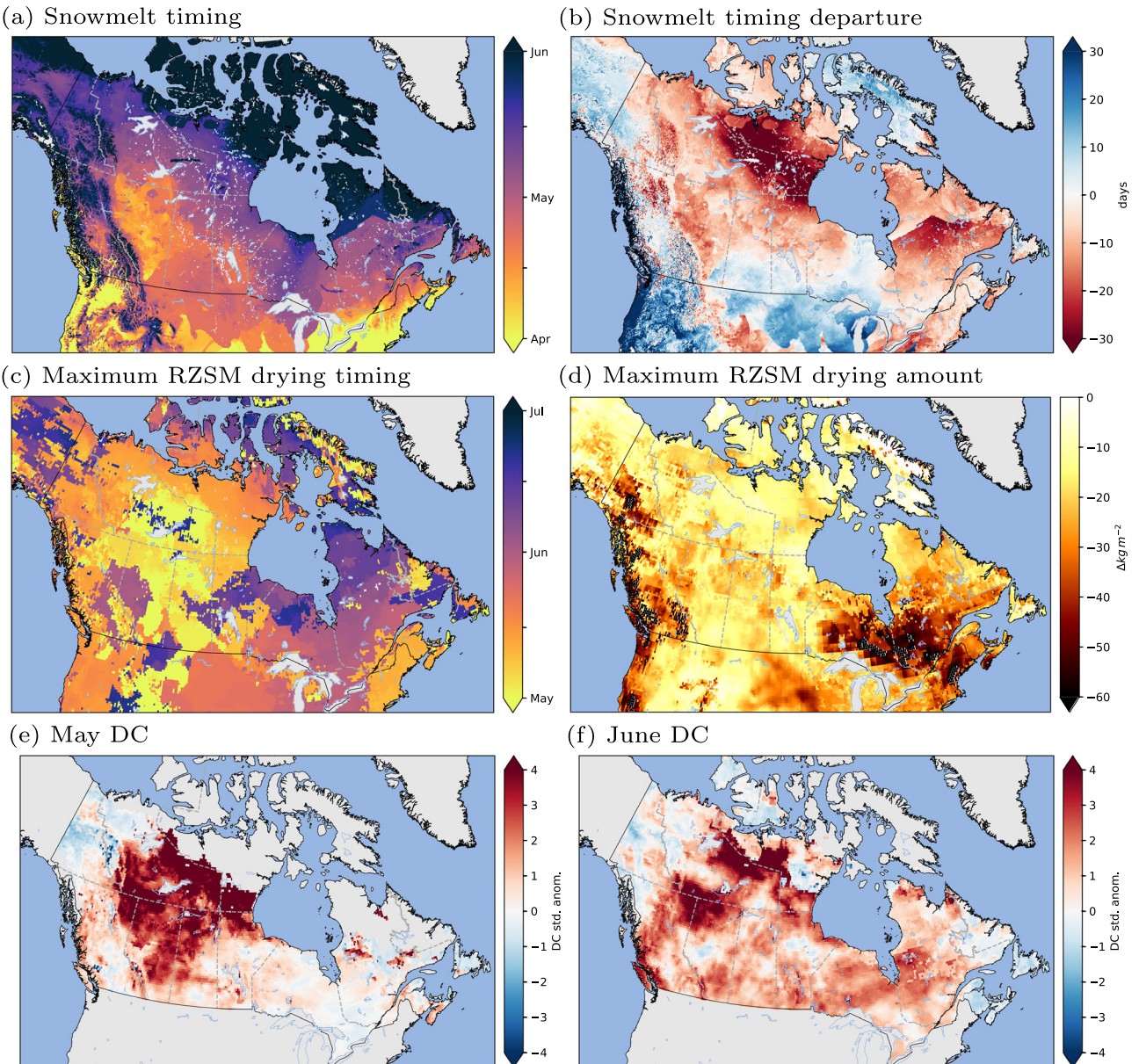

**Fig. 3 | Early 2023 fire season conditions for Canada. a** timing of snow melt; **b** snow melt timing departure (days) from historical average (2004–2022); **c** mid-point timing of maximum Root Zone Soil Moisture (RZSM) drying in any two-week period between May and July; **d** the corresponding maximum RZSM drying amount; **e** standardized anomalies of the May Drought Code (DC); **f** standardized anomalies of the June DC. Data sources are described in methods.

the persistent and widespread extreme fire weather conditions were likely related to the presence of frequent blocking events. Whereas for the baseline period (1991–2020) the mean number of annual (April–October) blocking days in Canada is ~15, the number of blocking days during 2023 far exceeded this value in most parts of the country with a mean value of ~50 (Fig. S2).

**Fire size and duration**

Most wildfires in Canada's managed forest are contained or self-extinguished before they reach 200 ha in size; however, fires that survive and exceed 200 ha ("large fires") account for approximately 97% of the area burned[4,41]. In 2023, there were 834 large fires in Canada, more than 2.5 times the 1986–2022 average of 320 (Fig. S4), as well as a substantially higher frequency of very large fires > 50,000 ha (Fig. 7a). The contribution of very large fires in 2023 is noteworthy: 60 of them were responsible for 73% of the total area burned, compared to an average of 7 very large fires with 41% of the total area burned in the

historical period (Fig. 7a). An uncommon event during most years, wildfire complexes where two or more wildfires eventually burn together resulted in 6 of the 10 largest, long-burning wildfires on record (Figs. S6, S7).

How did so many fires become so large in 2023? The likelihood of fire growth in any time period depends on the probability of extreme weather favoring growth, as well as the absence of fire-ending events such as significant rain or the onset of winter, assuming no fuel limitations[42]. Because the number of possible spread days for a given fire is limited by the fire duration, fire size also covaries with the time from ignition to control or extinguishment[43]. This is illustrated by the growth of the 10 largest wildfires that burned in Canada in 2023 (Figs. S6, S7). Of the 43 very large fires (>50,000 ha) that occurred in western Canada (west of 85 °W), the median duration was 82 days (90% CI: 32–156) in contrast to only 40 days (22–89) for the 17 very large fires that occurred in eastern Canada. The relationship between the number of extreme fire weather days (FWI > $FWI_{95}$) and fire size is shown in

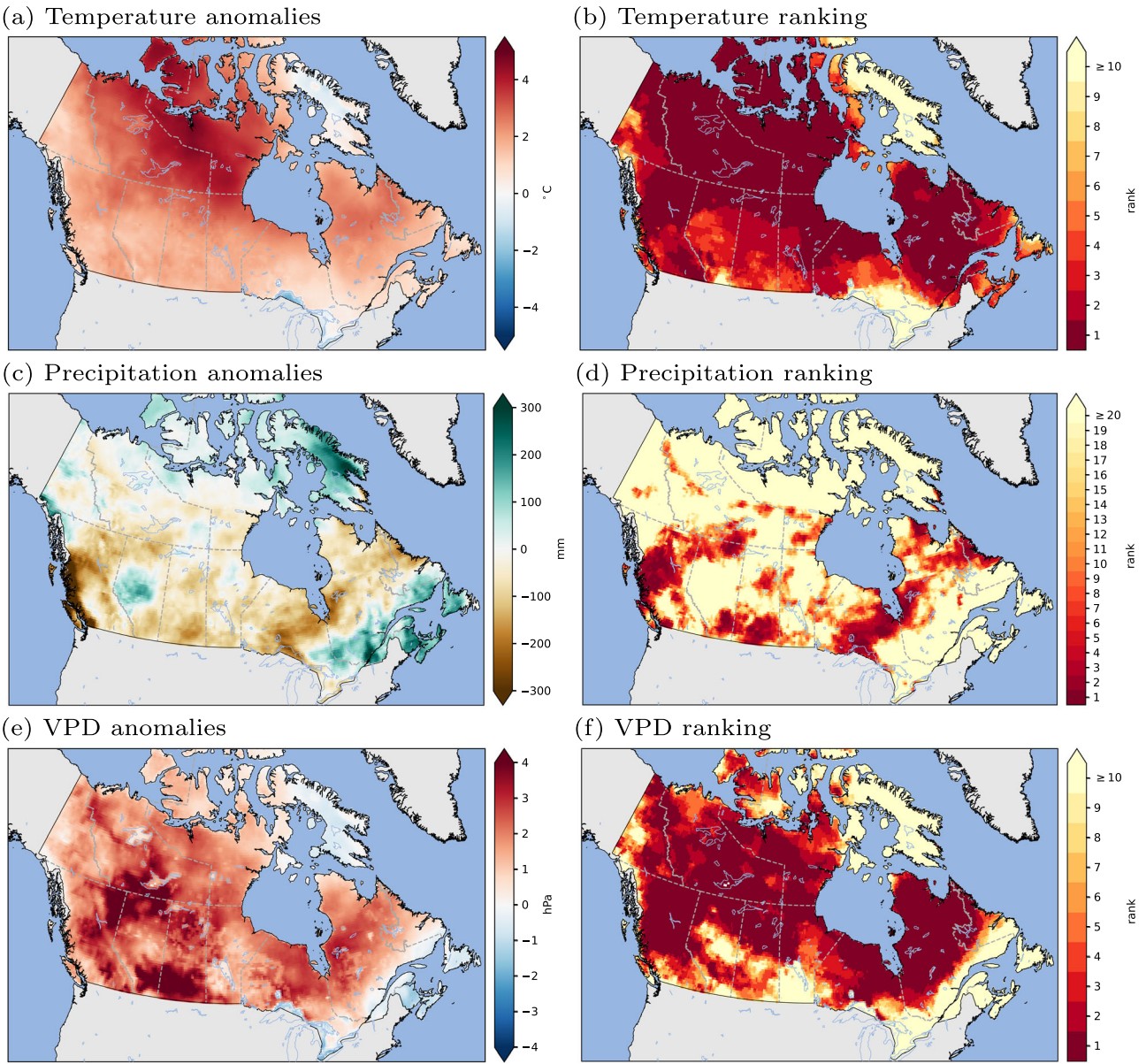

**Fig. 4 | 2023 fire season weather anomalies and rankings.** 2023 fire season (May–October) anomalies relative to baseline period (1991–2020) for: **a** 2-m temperature **c** total precipitation; **e** vapor pressure deficit (VPD); 2023 fire season (May–October) mean value ranking during period 1940–2023 for **b** 2-m temperature; **d**, total precipitation; and **f** VPD. Data derived from the ERA5 reanalysis.

Fig. 7b. In fact, a robust power-law relationship exists between extreme fire weather days (i.e., potential spread days) and fire size throughout Canada[10]. Interestingly, large fires in western Canada burned over a longer period than those in eastern Canada, but under similar numbers of extreme fire weather days, relative to the local FWI climatology. Although conditions were extreme in each regional context, typical fire weather conditions in western Canada are more severe overall due to climatologically higher aridity. The fires in eastern Canada that grew very large over a relatively short duration, compared to those in western Canada, may also reflect landscape factors such as fuel continuity and topography although further work is required to understand these regional differences[32].

The seasonal timing of ignitions also influences fire size, and this was particularly evident in 2023, as environmental conditions favoring fire growth over extinguishment persisted for extended periods in many regions of Canada (Figs. S6 and S7), particularly in western Canada where some fires burned for five months. Wildfires that began in the spring and were not extinguished

quickly grew larger (Fig. 7c) and accounted for an outsized proportion of the total burn area. The proportions of fires >1000 ha ignited in May, June, July, and August corresponded to 0.12, 0.24, 0.37, 0.08, respectively, but were responsible for substantially different proportions of the total area burned (0.35, 0.34, 0.24, 0.05, respectively; see Fig. 7d). A smaller number of large fires that began in April and September had a negligible contribution to the area burned nationally.

**Fire management response**
In 2023, approximately 79% of wildfires detected in Canada received full suppression response, with the remaining 21% receiving modified or monitored response (i.e., no direct intervention). However, modified and monitored fires contributed 61% percent of the total area burned. The number of active wildfires placed an overwhelming demand on resources throughout most of the 2023 fire season (Fig. S4). A national preparedness level of 5, signifying an extreme fire load, insufficient national resources, and limited exchange capabilities,

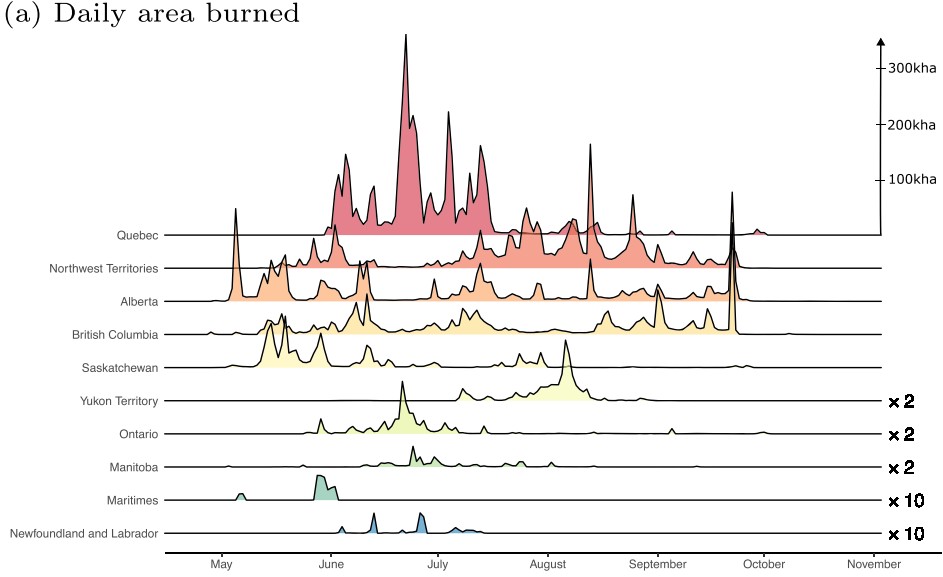

(a) Daily area burned

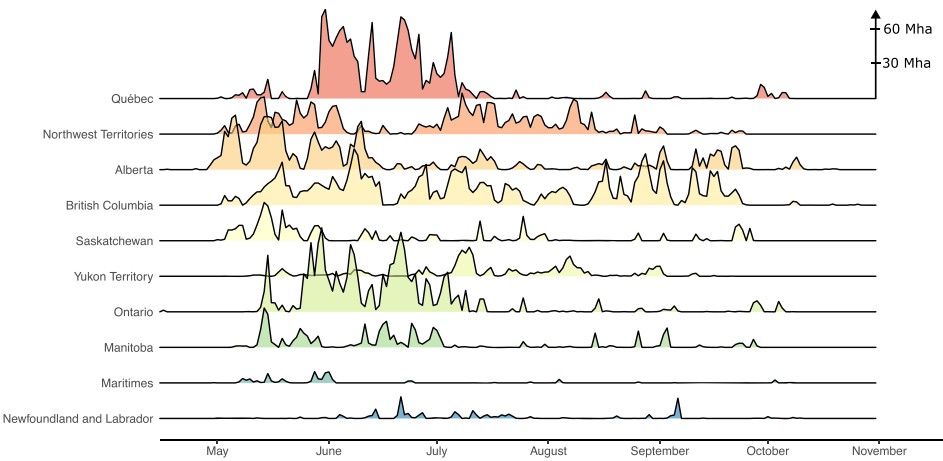

(b) Daily area with extreme fire weather

**Fig. 5 | Daily area burned and extent of extreme fire weather during 2023.**
**a** Daily area burned in 2023, by province/territory estimated from interpolated day of burn for fires ≥ 500 ha; scale was exaggerated (e.g., × 2) in areas with comparatively low area burned; and **b** the daily forested area in each province/territory with FWI values exceeding the 95th percentile of FWI (during the fire season). Nunavut is excluded from the plots due to its low total area burned and low forest cover.

was declared on May 11 and persisted for a record 120 consecutive days. Unprecedented personnel imports and exchanges occurred within Canada (over 1700 individuals), and over 5500 individuals from 12 countries and the European Union provided assistance. Additionally, the collective fire-fighting effort included contract firefighting crews, structural firefighters from municipal and local governments, individuals who volunteered, and the Canadian Armed Forces that were deployed to aid in firefighting efforts in multiple provinces. A significant number of firebreaks were created to halt or slow fire spread and protect communities and critical infrastructure. As a result of this record fire-suppression effort, approximately 85% of full-response fires were contained to under 200 hectares, in contrast to 62% of modified- and monitored-response fires that did not exceed this size.

**Societal and ecosystem impacts**
The human dimensions of wildfire also include the vulnerability of communities or individuals to wildfire smoke, evacuations, the loss of homes and structures, and in the worst-case scenarios, human lives[44].

Wildfires in Canada in 2023 resulted in profound societal impacts. Hundreds of thousands of people in communities were affected by evacuations, structure losses, power outages, and business interruptions, and millions of people in North America were exposed to wildfire smoke. Tragically, eight people working on wildfires in Canada were killed during the 2023 fire season. The approximately 15 Mha burned represents a vast area of fire-affected forest, with important implications for post-fire ecological recovery, habitat conservation, human use and enjoyment of forest resources, and the Canadian forestry industry.

Fine particulate matter with an aerodynamic diameter < 2.5 μm ($PM_{2.5}$) is a known air pollutant; exposure to $PM_{2.5}$ from wildfires has been shown to significantly impact human health[45]. For communities near sources of wildfires, smoke causes direct visibility degradation and hazardous exposures to air pollutants. Smoke can be transported hundreds to thousands of kilometers downwind, thus causing population-level exposure to elevated levels of $PM_{2.5}$ across large regions. Figure 8c shows the air quality warnings (bulletins) issued by forecasters at Environment and Climate Change Canada (ECCC) for the

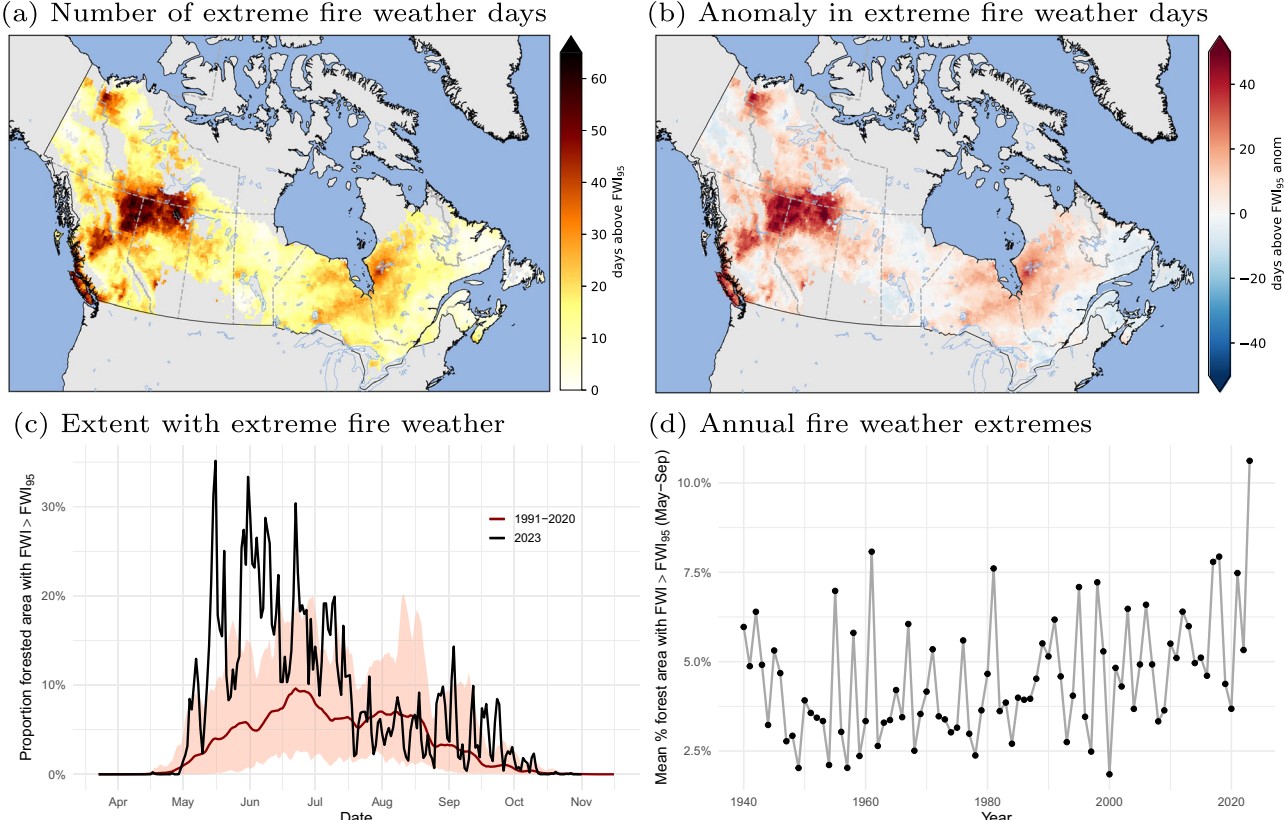

**Fig. 6 | Extreme fire weather conditions during 2023. a** Total number of days exceeding local 95th percentile of FWI for 2023; **b** Anomaly in total number of days exceeding local 95th percentile of FWI for 2023 relative to baseline period (1991–2020); **c** Proportion forested area exceeding local 95th percentile of FWI (black) compared with mean (red) values for 1991–2020 (confidence intervals shaded for 5th and 95th percentiles); **d** Mean proportion forested area exceeding local 95th percentile of FWI during fire season (May–September) for each year (1940–2023).

possible onset of poor air quality conditions[46] (Air Quality Health Index (AQHI)) for years between 2017 and 2023. Within the current decade, most AQHI alerts are the result of wildfire events across the country, with peaks from June to September. Compared to the annual average of about ~1300 alerts from 2017–2022, 2023 stands out with ~5000 alerts issued (Jan–Nov), due to the extremely poor air quality conditions from summer wildfires across Canada (see also Table S3). Fires in the southern Northwest Territories and northwestern Quebec were particularly influential sources of smoke in 2023 (Fig. 8a). In some regions, almost half of the summer (> 60 days) was spent in unsafe air quality conditions, sometimes in air containing more than 18 times the level of PM$_{2.5}$ required to trigger an air quality warning (Fig. 8b). Smoke

transport from fires as point sources (Fig. 8b) also affected many large communities along several eastern states of the USA. Furthermore, satellite imagery from early June 2023 also showed smoke from Quebec fires transported across the Atlantic, impacting countries in western Europe[2]. In 2023, Canadians experienced approximately eight days of poor air quality on average during the fire season; however, severely affected regions with lower populations, such as the Northwest Territories, endured up to 44 poor air quality days (Tables S5 & S6).

Most of Canada's population of ~41 million is concentrated in the southern part of the country, where fire activity is less prevalent. However approximately 12.3% of Canadians live within or adjacent to forested areas, in the wildland-urban interface[47], and these communities are more likely to experience an evacuation when threatened by fire. Indigenous communities, of which 32.1% of the on-reserve population is located within the wildland-urban interface, are especially vulnerable to evacuation[24,48]. Evacuations are extremely disruptive for community cohesion and economics, and have negative physical and mental health impacts[49,50].

In 2023, wildfires triggered evacuations of communities in 12 of the country's 13 provinces and territories[51]. Preliminary estimates indicate that approximately 232,000 people were evacuated in 282 events; the most evacuees of any fire season since records began in 1980[51], and nearly three times as many people as were evacuated due to the Horse River wildfire that burned through Fort McMurray in 2016. Five of the 10 largest evacuations ever recorded in Canada occurred this year (Table S4), with some communities such as Edson in Alberta and Lebel-sur-Quévillon in Quebec experiencing repeated evacuations. The 2023 fire season will undoubtedly have lasting social and health impacts throughout the country.

**Table 1 | Spearman correlation coefficient between daily forested area with FWI > FWI$_{95}$ and daily area burned, for each region in Canada**

| Region | Correlation |
|---|---|
| Quebec | 0.69 |
| Northwest Territories | 0.81 |
| Alberta | 0.85 |
| British Columbia | 0.91 |
| Saskatchewan | 0.81 |
| Ontario | 0.71 |
| Yukon Territory | 0.72 |
| Manitoba | 0.55 |
| Newfoundland and Labrador | 0.41 |
| Maritimes | 0.43 |

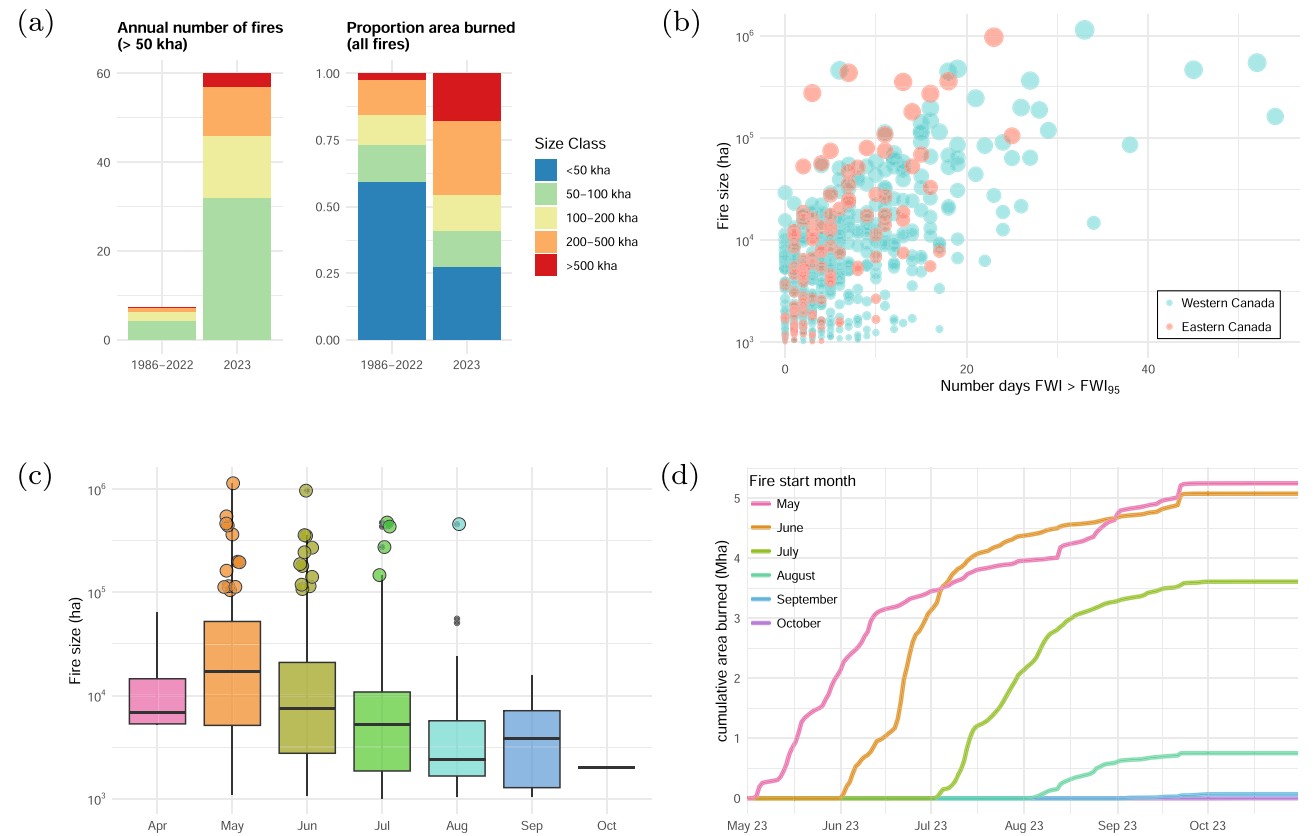

**Fig. 7 | Fire size and timing during the 2023 fire season. a** Left: number of fires > 50k ha for each fire size class for historical period (1986–2022, annual mean) and for 2023; Right: proportion of total area burned for each fire size class for 1986–2022 and 2023; **b** Fire size as function of potential spread days (extreme fire weather days; FWI$_{95}$) for western Canada (west of −85 deg longitude) and eastern Canada (east of −85 degrees longitude). **c** Distribution of fire sizes by start month for all fires > 1000 ha. (colored outliers correspond to fires > 100,000 ha). **d** Cumulative daily area burned for fires starting in each month between May–October.

Most Canadian ecosystems are adapted to large, intense wildfires, but their resilience may be compromised if wildfires are too frequent. Although recent wildfire activity has remained within the historical range of variability[52], the scale of this wildfire season ( ~ 4% of the total forest area burned) in Canada is well outside what was observed in recent decades. As such, the 2023 wildfire season will significantly alter forest landscapes. Nationally, wildfires predominantly burned forests (87.5%), except in ecotonal areas that transitioned to grassland ecosystems in the south (i.e., herbs vegetation type) and those that transitioned to tundra in the north (i.e., herbs and shrub types) (Table S2). Wildfires chiefly burned conifer and mixedwood forests, but a substantial area of broadleaf forests burned in Alberta due to this vegetation type regional dominance and the propensity of springtime, pre-greenup burning[53].

The 2023 fire season notably caused widespread burning of young forests (e.g, <30 years since fire or harvest), with over 1 Mha affected. This disturbance has the potential to cause extensive post-fire tree regeneration failures, because immature trees cannot provide enough seeds following a fire[54,55]. For instance, between 300 and 400 kha of forests might suffer post-fire regeneration failures in Quebec's commercial forests alone[56,57]. These failures, compounded by logging legacies, drought, and insect outbreaks, could reduce forest productivity[58] and carbon stocks[59],and accelerate the transition from boreal forests to open taiga, prairies or parklands[60–62]. Forest landscape changes from the 2023 wildfire season will have profound effects on forest ecosystem processes and biodiversity, with species adapted to early-stage or open-canopy forests benefitting, whereas those reliant on mature or old-growth forests being most negatively

affected in the near term[63,64]. The cumulative impacts of the area burned in 2023 coupled with the extensive anthropogenic disturbance legacies on the landscape will challenge the resilience of forest ecosystems, especially if fire activity continues to increase, as projected[65,66].

**Future outlook**
The 2023 Canadian fire season shattered national and multiple regional records, causing profound societal and health impacts. The combination of extreme weather and an extended fire season led to longer-lasting, hence larger, wildfires, resulting in a burned area approximately seven times the national average. Unprecedented synchronous fire activity strained fire management and response capacities across the country. Anthropogenic climate change facilitated the 2023 fire season, which is consistent with expectations of global warming put forward over the past several decades[6]. Disentangling the relative influence of human landscape legacies and worsening fire weather[8,18,67] on the current Canadian fire environment is ongoing, but it is inescapable that extreme heat and moisture deficits enabled the record-breaking 2023 fire season. The disproportionate effect a few days of extreme weather can have on the total area burned is also evident in this fire season[68,69], leading to worrisome prospects given projected future conditions[10]. Fire-fuel feedbacks may offset some potential for future fire activity[70,71]; however, increasingly severe weather overwhelms this resistance to burning[33]. Although this trajectory cannot easily be changed, we are not without options. Increased focus on innovative and integrated fire management strategies that prioritize risk mitigation are gaining momentum. This will require all

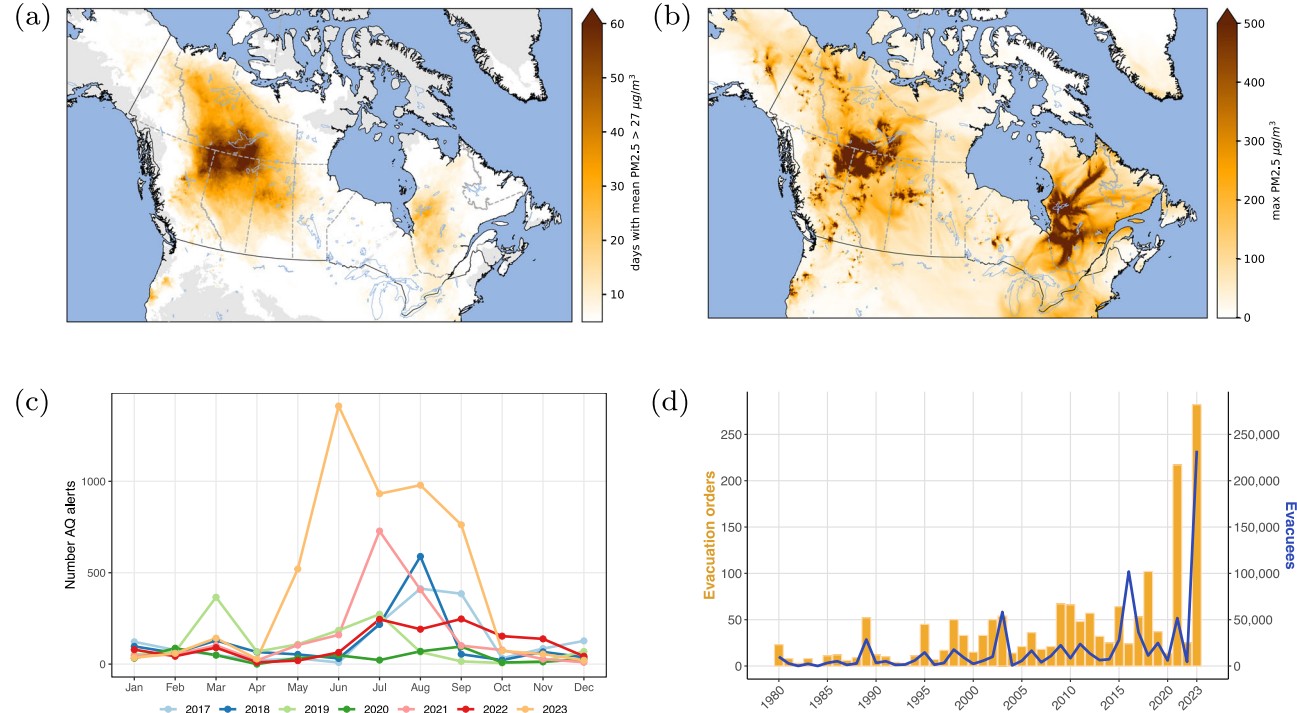

**Fig. 8 | Wildfire smoke and evacuation impacts for 2023. a** The number of days in 2023 with daily mean PM2.5 > 27 µgm⁻³; **b** the maximum daily value of PM2.5 that occurred during 2023; **c** the number of air quality bulletins each month issued by Environment and Climate Change Canada between 2017 and 2023 (Health and Air Quality Forecast Services Program, MSC-ECCC, December 2023); **d** The number of annual evacuation orders and evacuees from 1980–2023 using data from the Canadian wildfire evacuation database.

stakeholders to engage in fostering more fire resilient environments, as the 2023 fire season heralds the emergence of severe climate change impacts on fire activity decades earlier than previously anticipated.

## Methods

### Wildfires in Canada

Wildfires have been a longstanding ecological process in many ecosystems of Canada since the end of the last glaciation. Fire-prone regions of Canada have predominantly boreal and temperate continental climates, with the majority of wildfires occurring between April 1 and September 30. Even though about half are ignited by humans, lightning-caused fires are responsible for a greater proportion (~90%) of the area burned[4], occurring in remote areas[72]. In some parts of the country, Indigenous cultural burning during spring or fall also drove local fire regimes, though these practices were suppressed beginning in the early 20th century[73,74]. From a meteorological standpoint, the Canadian wildfire season is typically characterized by a series of high-pressure atmospheric systems resulting in drying periods of several days to more than one week, interspersed with rain events[75]. Lightning storms occurring during or after these dry spells can cause several hundred fire starts within a few days in a given region. When followed by persistent warm and dry weather and strong winds, extensive forested areas can burn as crown fires[76].

### Fire data

In Canada, high-quality spatial wildfire perimeters are available from the National Burned Area Composite (NBAC)[1]. At the time of writing, the NBAC product had not yet been completed and consolidated with agency data, so we considered a variety of provisional fire data sources to form a complete picture of fire activity for 2023.

### Agency fire data

During the fire season in Canada, fire information reported by fire management agencies are collated by the Canadian Interagency Forest Fire Centre (CIFFC). These data include reported date, fire ignition cause, and fire management response type. We downloaded reported data for 2023 (https://ciffc.net/, accessed December 29, 2023) to determine the number of incidents, management responses, and fire cause.

### Subdaily fire detections

Near-real-time fire perimeter polygons were obtained from the M3 fire perimeters[77] derived from thermal anomaly (hotspot) point detections using a two-step buffering process as follows. First, VIIRS (375 m) hotspots (VIIRS and MODIS (1 km) in Quebec) are buffered out $r_{out}$ meters to form circular polygons. Boundaries between intersecting circular polygons are dissolved to form larger polygons. Second, perimeters of the resulting polygons are buffered in (contracted) $r_{in}$ meters. The choice of $r_{out}$ and $r_{in}$ for each ecozone were determined by comparing buffered polygons from 2012–2021 with higher-resolution fire perimeters from the NBAC. Selected radii maximize the intersecting area, and minimize commission and omission errors.

### National Burned Area Composite

The provisional 2023 NBAC fire perimeters were created using cloud and shadow-free satellite imagery from the Google Earth Engine Collection-2 catalog (Landsat-8 and -9 data combined at 30-meters resolution, and Sentinel-2 data at 20-meters). We produced post-fire mosaics using median compositing of images collected over a 30-day period starting after the last hotspot acquisition date of the corresponding M3 perimeter. A pre-fire mosaic was derived likewise using the dates of the post-fire mosaic one year earlier. Image compositing was performed independently for Landsat and Sentinel, and an analyst selected the best quality pre- and post-fire mosaics for image mapping. In this procedure, an adaptive threshold was applied based on the differenced Normalized Burn Ratio to map pixels corresponding to burned areas[1]. The largest fires contributing the greatest area burned were targeted for NBAC mapping.

## NBAC-M3 2023 hybrid dataset

We produced a hybrid fire perimeter dataset ("NBAC-M3") using the combination of all preliminary NBAC fire perimeters and, for the 18% of the 2023 area burned which was not yet mapped, M3 perimeters[78]. We removed water bodies using the CanVec hydrographic features polygons[79]. It was necessary to correct the apparent burned area from M3 buffered hotspots, since buffered points typically overestimate area burned due to the presence of unburned islands and water bodies that are not accounted for in the buffering process. We did this by dividing the NBAC area burned by the total area burned reported in historical M3 buffered hotspots for 2012–2022, stratified by ecozone and year, thereby producing ecozone-specific calibration factors. We multiplied the calculated area of the M3 polygons for 2023 by the correction factors, within ecozones. Finally, polygons less than 1 ha in size were removed from the final dataset to minimize any introduced artifacts from the above process. We then summed the calibrated M3 area burned, with the mapped area burned from NBAC polygons to estimate the total area burned in 2023.

We calculated the proportion of land cover type that burned by intersecting the final NBAC-M3 dataset with the SCANFI classified land cover raster dataset[80], excluding nonfuel. This dataset uses National Forest Inventory plots and satellite data to classify land cover into Water, Rock, Bryoid, Herbs, Shrub, Treed broadleaf, Treed mixed and Treed conifer land cover classes. Proportion land cover type burned, by province or territory, is reported in Table S2.

## Daily fire growth

To estimate daily fire growth, we interpolated the fire detection date from hotspots for all wildfires in the NBAC-M3 dataset with a final area over 500 ha. Detection date[77] was corrected to local standard time and interpolated using ordinary kriging to a 180-m grid, following Parks[81]. A filter was applied to detections prior to April 1st to remove false detections outside of the main fire season. In total, 641 fires were interpolated, accounting for ~99.5% of the total area burned in 2023.

## Snowmelt timing

Daily snow cover data were obtained from the Interactive Multi-sensor Snow and Ice Mapping System produced by the United States National Ice Center (USNIC)[82] at 4-km resolution. We defined snowmelt timing at each location as the first day of the longest snow free period at that location for any given year. In addition, snow melt timing departures for 2023 were determined by the anomaly (days) of the 2023 snowmelt timing from the mean timing of the period 2004–2022.

## Drought data

The root zone soil moisture is an indicator of the moisture content of the top 1 m of soil, and as such, is a useful measure of drought. We obtained the root zone soil moisture from the Global Land Data Assimilation System (GLDAS-2.2)[83], a land surface model constrained by Data Assimilation of the Gravity Recovery and Climate Experiment. The GLDAS-2.2. data are daily at 0.25° resolution, and were subset to North America. To determine rapid onset of drought, we calculated the maximum amount of drying (i.e., reduction in root zone soil moisture values, kg m$^{-2}$) that occurred using a 14-day sliding window from May 1st to June 30th. The timing of rapid onset was defined as the midpoint of the window corresponding to the maximum 14-day drying.

## Weather and fire weather

Weather data were obtained from the ERA5 reanalysis via the Copernicus Climate Change Service[84]. ERA5 includes an extensive set of surface and upper air atmospheric variables, available hourly and globally at 0.25° resolution. We downloaded 2-m temperature, 2-m dewpoint temperature, precipitation, and the 10-m U and V components of wind. Relative humidity and vapor pressure deficit (VPD) were estimated from temperature and dewpoint temperature, using the equation from Alduchov and Eskridge for VPD[85].

We examined mean daily anomalies of temperature, precipitation and VPD averaged between May 1st and October 31st (fire season) of 2023 compared with the baseline period (1991–2020); we also plotted the rank of the fire season mean of each variable for 2023 compared with the period 1940–2023.

The Canadian Fire Weather Index System (CFWIS) − part of the Canadian Forest Fire Danger Rating System[34] − is an empirical model that combines surface temperature, relative humidity, 10-m open wind speed, and 24-hour accumulated precipitation collected at solar noon[34]. The System outputs represent proxies to fuel moisture in the forest floor, potential fire behavior and an overall fire danger rating (the Fire Weather Index, FWI). We calculated CFWIS outputs from ERA5 weather following a procedure to account for interseasonal drought conditions[86]. Fire season was determined using the startup condition that daily maximum temperature should exceed 12 °C for three consecutive days, and annual shutdown conditions when temperatures fall below 5 °C for three consecutive days, serving as a proxy for snow free periods[7].

We calculated a climatology of extreme FWI values at each ERA5 grid cell, defined as the 95th percentile of FWI values ($FWI_{95}$) between 1991–2020 (omitting values outside of the fire season). An extreme fire weather day is then any day for which $FWI > FWI_{95}$. Using percentiles to define extreme fire weather accounts for the fact that the fire weather relationship (e.g., fire ignition and spread) are heterogeneous across the landscape. The extent of extreme fire weather conditions was further defined as the forested area exceeding the local 95th percentile of the $FWI$[87,88] ($FWI_{95}$, defined by the fire season period during 1991–2020). We defined forested area by applying a threshold of > 20% canopy cover to the NASA Making Earth System Data Records for Use in Research Environments (MEaSUREs) Vegetation Continuous Fields (VCF) Version 1 data product[89] (VCF5KYR).

To examine the influence of large-scale atmospheric patterns over Canada in 2023−and in particular, atmospheric blocking events−we applied the identification algorithm for persistent positive anomalies in 500-hPa geopotential heights, previously developed by Sharma et al.[40]. A persistent positive anomaly was identified for contiguous regions where the 500-hPa geopotential heights exceeded one standard deviation above the local seasonal climatological mean (for the baseline period 1991–2020) and persisted for at least 5 days, with the additional criterion that each event reached at least 100,000 km$^2$ during its evolution. We applied the algorithm using a domain that covers the entirety of the forested area of Canada. We considered the sum of days with a persistent positive anomaly present as a proxy for the number of blocking days at each location.

## Air quality

We compiled PM2.5 concentration maps from the first 24-hour model outputs of the ECCC operational Regional Air Quality Deterministic Prediction System-FireWork[90]. FireWork is a state-of-science numerical air quality forecast system with near-real-time wildfire emissions[89,91]. The core of the FireWork system is the GEM-MACH meteorology-chemical-coupled model that simulates the dynamics of chemical composition, accounting for detailed gas-, aqueous- and particle-phase chemical reactions and atmospheric physics. The model domain covers North America with a 10-km grid resolution. Hourly wildfire emissions were estimated using the Canadian Forest Fire Emissions Prediction System (CFFEPS)[89] for fire grid locations representing hotspot areas (past-24 hr satellite detection by VIIRS sensors). We estimated emissions with forecast meteorology, fuel type, and burn areas through the NRCan Canadian Wildland Fire Information System[92]. Output of the FireWork system consists of hourly concentrations of gas-phase and aerosol species. The total fine PM components represent simulated PM2.5 concentrations. We adopted thresholds for

ambient levels of $PM_{2.5}$ as regulated under the Canadian Ambient Air Quality Standards (CAAQS, https://ccme.ca/en/air-quality-report), with daily and annual standards determined as a 24-hour average concentration of 27 µg m$^{-3}$ and an annual average of 8.8 µg m$^{-3}$.

Air Quality (AQ) Alert Bulletins were issued by MSC-ECCC, 2017-2023. Data represents the number of all AQ Alert bulletins issued throughout the period. Each AQ alert has a minimum of 2 bulletins issued (issuance and termination), plus a number of continued bulletins based on the persistence of the event. Each AQ Alert's area of coverage varies in size based on the geographical extent of the AQ event.

### Evacuations

Evacuation data for Canada for 1980–2023 were obtained by request from the Canadian Wildland Fire Evacuation Database[51]. These data are compiled using an exhaustive search of media reports and quality control measures as described by Tepley et al.[24]. Evacuation data from 2023 are provisional and are subject to updates as evacuation numbers are confirmed.

## Data availability

Supporting data for this paper are available at the Centre for Open Science OSF data repository (https://doi.org/10.17605/OSF.IO/NY2R4). Snow cover data were obtained from the United States National Ice Center, https://nsidc.org/data/g02156/versions/1. Data from the North American Drought Monitor was downloaded from: https://droughtmonitor.unl.edu/NADM/Statistics.aspx .GLDAS-2.2 data were obtained from: https://disc.gsfc.nasa.gov/datasets/GLDAS_CLSM025_DA1_D_EP_2.2/summary .ERA5 reanalysis data were obtained from the Copernicus Climate Change Service (2023): ERA5 hourly data on single levels from 1940 to present. Copernicus Climate Change Service (C3S) Climate Data Store (CDS), https://doi.org/10.24381/cds.adbb2d47 (Accessed on 07-Jan-2024).We acknowledge the use of data and/or imagery from NASA's Fire Information for Resource Management System (FIRMS) (https://earthdata.nasa.gov/firms), part of NASA's Earth Observing System Data and Information System (EOSDIS).

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

## Acknowledgements
The authors would like to thank Dawn McVittie (Canadian Forest Service) for compiling the 2023 evacuation data, Melissa MacDonald at Environment and Climate Change Canada for providing the air quality alert data, and Claudia Castillo for graphic design. We would also like to thank David Peterson, Michael Fromm and Rene Servranckx for sharing pyro-cumulonimbus event statistics for 2023.

## Author contributions
P.J., Q.E.B, S.T., E.W., and M.A.P. designed research; P.J., Q.E.B., D.C.A., J.C., P.E., J.L., K.M., and R.S.S. curated datasets and performed analysis; P.J. and Q.E.B. created visualizations; P.J., Q.E.B., E.W., M.A.P., and S.T. led the writing and editing of the manuscript; and D.C.A., Y.B., R.D.C., J.C., P.E., M.F., M.P.G., C.C.H, J.L., K.M., R.S.S., D.K.T., and X.W. contributed writing, and conducted reviewing and editing.

## Competing interests
The authors declare no competing interests.
