## [Peer Review File · Nature Communications]

Drivers and Impacts of the Record-Breaking 2023 Wildfire Season in CanadaREVIEWER COMMENTS

Reviewer #1 (Remarks to the Author):

See attached PDF.

[Editorial Note: this PDF is displayed after the comments of Reviewer #2]

Reviewer #2 (Remarks to the Author):

Comments for the Manuscript NCOMMS-24-16816 “Canada Under Fire – Drivers and Impacts of the Record-Breaking 2023 1 Wildfire Season”

Summary:

The authors present an overall assessment of the record breaking 2023 fire season for Canada, providing information about the causes, the impacts, and the implications. Throughout the manuscript, authors provide quantitative information to support their analysis, based on several lines of evidence, datasets and literature. Authors provide a comprehensive overview of the wildfire season, with regional statistics referring to the burned area and the drivers that led to it. They effectively summarize the drought and fire weather conditions that led to the exceptional fire year, and present considerations referring to the fire management intensity and its implications to the burned area. Finally they consider socioeconomic impacts of the fire season. The manuscript is well written and well organized, while the figures (except for some minor comments provided here) are of great quality.

Main comments:

While there is not much to criticize in this work on what is already in the manuscript, what seems to be missing is a good insight into the vegetation that was burned, i.e. types of trees/forest per region, as well as the vegetation conditions prior the wildfire events compared to the climatology.

Information about the types of vegetation may also partly explain the regional differences between the Eastern and Western Canada. Regarding the vegetation conditions, while the DC code of the FWI partly covers this aspect, the DC is a very simple soil moisture accounting scheme and hence can provide an estimate on the “agricultural” drought status, but not the vegetation – despite the fact that the FWI was initially created for Canadian wildfires. A suggestion would be to expand their analysis into a remotely sensed index (like NDVI if there is not any other better?) as a comparison between the climatological values and the 2023 values.

Subchapter “Fire size and duration”: Authors define the potential spread days as the days where the FWI exceeds the 95th percentile of May-October climatology. But is this true? To what I can interpret from Fig S6, in many cases the fire sustains (while not flourishing) for long periods (in days) with pyro-meteorological conditions below 95th percentile. So the question that arises is whether

there is a rough threshold in the FWI that can be considered as a "fire-ending event"?

Other comments

L224: Figure 5 does not seem to provide any correlation information between FWI and BA. There is obviously a covariation in both quantities. It would be interesting to also provide some type of correlation (Pearson or Spearman maybe?) . As also stated in the manuscript, Fire weather index can correlate well with the logarithm of the BA.

In figure 3a, the snowmelt timing pattern in the central Quebec region, shows a rapid transition between May and ~Jun values. Is there any elevation/aspect/soil/climate reason for that? The same reason also causes the same pattern in Figure 3b, and for some reason partly in the Figure 3f (June DC) but interestingly not in the Figure 3e (May DC)? Authors should check for potential issues in the data or their processing.

Figure 4: Authors should add titles in the sub-figures of Figure 4 as they have done in Figure 3. Also Figure 4 caption does not make any sense, something is going wrong.

Figure 5: What is the x2 and x10 values mentioned at the right of Figure 5a? This should be mentioned in the figure caption. Also, is there any way to include y-axis information?

Review of Jain et al., “Canada Under Fire...”

Philip Higuera, Missoula, MT, 13 April, 2024

This is an excellent contribution that I was excited to review. Given the record-breaking nature of Canada’s 2023 fire season, there will no doubt be broad interest in this work, and it has the potential to be widely cited by scientists, managers, and policy makers. The data and analyses are simple and sound, providing a critical context for how society makes sense of the 2023 fire season in Canada. This is critical work.

Given the importance of the topic and the likely broad interest, the bulk of my feedback is intended to help clarify and simplify messages and make this easily digestible by non-specialists. In my experiences with *Nature Communications* and other high-profile journals (as a reader, reviewer, and [co-]author), the reach of the work will be significantly improved with simple language. This differs from if this paper were published in a fire-specific journals (e.g., IJWF). This work could be taken up by researchers from multiple fields, journalists, and policy makers, so there is great opportunity to provide a clear message to those audiences.

1. **Use numbers whenever possible and be as precise as possible.** For example, consider replacing superlatives like “shattering” and “eclipsed” or subjective qualifiers like “many more” and instead use numbers, e.g., “X-times larger than” or “more than X-time higher than...” Use the hard-earned numbers produced by this research. Suggestions are below.
2. **I kept waiting for a clear statement on climate and climate change, and text that explicitly helps the reader fit 2023 into the broader context of global warming and shifting fire regimes.** This is where this work will end up, regardless of the authors’ intentions. The statistics are impressive, and one might think the statistics could “speak for themselves”, but the authors have an opportunity to be explicit with how the audience should integrates 2023 in the context of topics like “wildfire crisis,” “climate change,” “human development”, and “forest management.” Connect the dots for the readers.

It was not until the last sentence of the paper that a critically important statement is made: “...*the 2023 fire season may foreshadow the emergence of severe climate[-]change impacts on fire activity decades earlier than previously anticipated.*” This seems like a very relevant thesis for the paper itself, but it’s not a message that comes through clearly throughout the paper, particularly the “earlier than previously anticipated” point. That has huge implications. Overall, there are several paragraphs where “the lead is buried” a bit, noted below.

3. The data presented clearly support the statement that “extreme climate enabled the record-setting 2023 fire season.” Is that in fact a main point of this work? Full disclosure, I am a fan of the term “climate enabled” (e.g., Higuera and Abatzoglou 2021, GCB). The term clearly attributes climate as a critical contributor, but it does not hang everything on climate, since (as the authors know and communicate well) ignition and extreme fire weather are also needed. All elements are well documents in the manuscript. I encourage the use of “enabled” in several parts of the paper, detailed below.

4. The paper lacks the important context that “fire is a longstanding component of” Canadian forest ecosystems, for millennia. This is simple, but without this, many readers could walk away thinking that 2023 was a crazy climate-change impact entirely. It was not; it was an expression of a longstanding natural phenomenon, under climate change (or “on steroids”).
5. Consider using the term “average” instead of “normal” – simply given the normative use of the term “normal” outside of climatology, statistics, and research in general. This will improve the way this is communicated to non-specialists, particularly journalists.
6. Limit parenthetical statements. If something is important, keep it in the sentence. Limit the use of acronyms, to keep things digestible for a broad audience. Suggestions below.
7. “Data” is used as a singular. Convention is that “data” is plural...but conventions change.

Detailed comments:

L 54: Consider flipping this sentence to start with “The 2023 wildfire season in Canada was unprecedented in its scale and intensity” – and, to clearly justify “unprecedented” consider simply adding “...scale, intensity, and human impacts”. It was 100% unprecedented in human impacts – no arguing about that part.

L 56: Consider clarifying “periods of dense smoke that caused public health concerns” – e.g., “thousands exposed to hazardous air quality [from wildfire smoke]...”

L 58: Take the actual statistic out of the parentheses here – this is a big number that will be cited. E.g., “The record-breaking fire season, with over 15 Mha burned, can be attributed to...”

*Here is the first opportunity to make a clear statement on attribution: both items listed under “can be attributed to” are climate-related: snowmelt and drought. In the authors’ view, was the 2023 fire season largely a function of climate? State that; otherwise, readers will wonder about fuels, forest management, human ignitions, etc.

It is true that the record-setting 2023 fire season was...enabled by extreme climate conditions, consistent with expectations of a warming planet? And, perhaps emerging earlier than expected under climate-change scenarios? If so, highlight that in the abstract.

L 61: Example of replacing “normal” with “average” and removing parenthetical statements: “...was a staggering 2.2 C warmer than the 1991-2020 average, enabling...” And here, these conditions enabled dry fuels – not really extreme fire weather per se, no? When I read “extreme fire weather,” I’m thinking about high winds, although I know the FWI integrates fuel moisture. It seems to me that drought (+) created extremely dry fuels, which then enabled successful ignitions and rapid fire spread when winds were high.

L 63: Suggestion: “...in the country and across North America” OR “...across North America.” As noted, smoke impacts on the NE USA impacted health and prompted much discussion.

L 70: "...(c. 1975) over two times the previous record of 6.7 Mha..."

L 80: The text would be improved with a transition between these first two paragraphs. The second paragraph is highlighting climate as an important contributor to the 2023 fire season, but this is backed into over a number of sentences. The transition from the impacts on communities to climate statistics is sharp. Consider something like: "The 2023 fire season was enabled by record-setting climate. CLIMATE STATISTICS..."

L 92: "entrained" here seems odd. Was this intended to be "enabled"? That would make sense.

L 94-95: This seems like the topic sentence (buried at the end of the paragraph), or at least an important topic: increasing wildfire activity in Canada as symptomatic of changing climates. Or, more specifically, symptomatic of global (and regional) warming.

L 103-105: Again, this seems like a clear topic sentence, hiding in the paragraph: "...2023...was unprecedented and...not anticipated until later in the century based on climate[-]change projections." This is a really critical point, worth highlighting and leading with.

L 113: For brevity, consider: "Here we provide an overview...and examine selected drivers and impacts..." Note, only selected driver, and esp. only selected impacts, are presented here. Consider stating the objective as completed contributions, vs. the "-ing" of each objective. Finally, it's worth emphasizing the importance of this work in helping contextualize what was an extraordinary event, not only for Canada, but globally. Documenting and contextualizing these impacts of global warming is a critical task more broadly.

L 148: "...since satellite records began in 19XX" – let the reader know when the records began.

L 150: The section title could be confusing, with "high" potentially read to refer to altitude. Consider: "Exceptionally large total area burned" or simply "Record-setting annual area burned."

L 153: Note that "total burned area" and "area burned" are both used. Either works, but being consistent will help for a general audience.

L 154: "...historical national average."

L 168: Consider: "...the total area burned (19XX-20XX average – 91%)" – to avoid the reader getting distracted by asking the question "over what period?"

L 182: Is it known, with the statistics on hand, what % of wildfires that resulted in home or structure loss originated from human ignitions? I asked from my own perspective and interests (e.g., Higuera et al. 2023, PNAS Nexus), but I know others would wonder this too. This is critical in the western US, as it clearly points to the fact that most of the "wildfire crisis" (i.e., home and structure loss) is attributable to human ignitions, whereas 88% of wildfires don't result in any home or structure loss. If not possible with the stats. on hand, no need to dig.

L 188: Consider "early snowmelt" vs. "premature"? i.e., objectively earlier than average.

L 197: "near-average"

L 209: "hotter and drier...than average"

L 210: "...warmer than the 1991-2020 average,..."

L 221: Consider: "Fire growth is largely determined by the coincidence of dry fuels, ignitions, and the occurrence of extreme weather conditions..." The text mixes "extreme weather" and "extreme fire weather." I know the latter inherently includes dry fuels, but to the general reader, "extreme weather" is most easily linked to things like high winds.

L 244: Consider spelling out PPA, since the acronym is only used once in this paragraph.

L 249: For brevity, consider keeping figure references in parentheses at the end of the sentence.

L 256: Consider: "In 2023, the 834 large fires in Canada was more than 2.5 times the 1987-2022 average of 230."

L 272-273: Isn't the definition of potential spread days imply "days with $FWI > FWI_{95}$ "? FWI_{95} was previously defined in the text.

L 280: Consider just deleting "bottom-up factors" and just state "may also reflect fuel continuity and topography" as "bottom-up factors" is a fairly fire-specific term.

L 294: Subsection title here is a little vague, with "considerations" Consider "response".

L 306-307: Not sure if this holds, but in the US we emphasize that fire/fuel breaks are intended to change fire behavior (including slower rate of spread), but not necessarily halt fire spread.

L 314: Strongly suggest deleting "Aside from fire management..." as this qualifier suggests a narrow view of fire impacts. It is well acknowledged that fire impacts – and specifically human impacts – go well beyond fire management personnel. Suggestions: "The human dimensions of wildfire include the vulnerability of communities or individuals to wildfire smoke, evaluations, and loss of homes and structures, and in the worst-case-scenarios human lives." The "even lives" makes it sounds like a rare possibility – it's happening more and more each year. In fact, it is remarkable – literally worth noting in the text – that the death toll from the 2023 fire season in Canada (not in the US) was kept so low.

Consider citing, in the first sentence of this paragraph, a paper on human dimensions of wildfires and vulnerability: "Social drivers of vulnerability to wildfire disasters: A review of the literature: <https://www.sciencedirect.com/science/article/pii/S0169204623001160>

L 332: Over 3x more?

L 381: For the ecological impacts of the 2023 wildfire season, consider citing Coop et al. (2020), for a broad, recent, North American overview: <https://academic.oup.com/bioscience/article/70/8/659/5859066>

L 387-391: This is all true, but generally a weak way to link the 2023 fire season to climate change, global warming. Aren't we past the stage where we wonder if we're seeing the impacts of global warming manifest through wildfire activity? Consider something like "Although future research is necessary to quantify the degree to which anthropogenic climate change facilitated the 2023 fire season, it is clear that extreme climate conditions, all consistent with expectations

of global warming,...” This is my opinion, but I think it's a disservice if this text comes off as wondering if climate change was a factor here; details and specifics, yes, but overall, this is consistent with what we've expected and predicted, since Flannigan and Van Wagner 1991 (!).

L 396: "...moisture deficits enabled the record-breaking 2023 fire season." This fits under the framework of "climate enabled, weather driven" as an explanation for extensive fire activity.

L 402: This suggestion of prioritizing fire prevention is just doubling down of fire suppression? This seems odd, not to mention very dangerous.

L 404-405: As note above...this is a HUGE and very important point. The paper would benefit if this was highlighted throughout, and expended upon a bit more.

L 409: "Wildfire in Canada" – this is at least one place where it should be noted that wildfire in Canada is a longstanding process, for millennia.

L 485: This page of method has a lot of acronyms. If the acronym is only used once or twice in a paragraph, consider eliminating it and just spelling out the term, for clarity.

L 519: I'm familiar with the CFFDRS...but I don't know what "screen-level temperature" are.

L 534: What size is the "grid-level" here?

Figures

Figure 1: Strongly suggest changing the text (in the upper-right corner) to "...largest extent of land *impacted by fire* in Canada's history" less you perpetuate the misconception that wildfires destroy all land they burn across. The land is not "consumed" – it's still there (thankfully)!

*For Fig. 1 and 2 – are these statistics for "total area" or just "forested area"? I think the former...but specifying would be helpful.

*Fig. 3: Suggest using "average" here too instead of "norm".

*Fig. 4: Including titles after the panel labels ("a," etc.) would be helpful here, as in Fig. 3.

*Fig. 5: Great figure! Is it possible to have a y-axis for each panel, at least for one of the subpanels? There's no ways gauge the value of area burned (a) or number of days (b) currently.

Fig. 6: Having panel (a) as an anomaly (relative to average) as well would be useful. It would arguably be more useful than panel (c) in this figure (which could go to the appendix). That would keep this as a four-panel figures.

Fig. 7: In my opinion, this entire figure could go to an appendix.

Fig. 8: Consider having panel (b) as an anomaly, instead of the PM2.5 values. That would help highlight what's unusual about 2023.

REVIEWER COMMENTS

Reviewer #1 (Remarks to the Author):

Philip Higuera, Missoula, MT, 13 April, 2024

This is an excellent contribution that I was excited to review. Given the record-breaking nature of Canada's 2023 fire season, there will no doubt be broad interest in this work, and it has the potential to be widely cited by scientists, managers, and policy makers. The data and analyses are simple and sound, providing a critical context for how society makes sense of the 2023 fire season in Canada. This is critical work.

Response: We thank the reviewer for their positive assessment of this paper. Their feedback has greatly improved the manuscript. Please see detailed responses to each of the comments below.

Given the importance of the topic and the likely broad interest, the bulk of my feedback is intended to help clarify and simplify messages and make this easily digestible by non-specialists. In my experiences with *Nature Communications* and other high-profile journals (as a reader, reviewer, and [co-]author), the reach of the work will be significantly improved with simple language. This differs from if this paper were published in a fire-specific journals (e.g., IJWF). This work could be taken up by researchers from multiple fields, journalists, and policy makers, so there is great opportunity to provide a clear message to those audiences.

1. Use numbers whenever possible and be as precise as possible. For example, consider replacing superlatives like “shattering” and “eclipsed” or subjective qualifiers like “many more” and instead use numbers, e.g., “X-times larger than” or “more than X-time higher than...” Use the hard-earned numbers produced by this research. Suggestions are below.

Response: We agree that some of the subjective language could be changed. However, where numbers are also given to back up these statements, we do not believe the superlative requires removal as it relates to a quantitative statement. For example, this is the case on Lines 69-71: “At approximately 15 Mha, the area burned was by far the highest since the start of comprehensive national reporting (c. 1972), shattering the previous record of 6.7 Mha in 1989” we have decided to keep the word “shattering” as the relevant numbers are also provided.

As suggested by Reviewer #1, we have made the following changes:

Line 60 - (abstract): removed ‘shattering’

Line 72 - replaced “a massive” with “one”

Line 171 - replaced “eclipsed” with “was more than double”.

Line 281 - we removed the expression “many more” and used the number of large fires directly.

2. I kept waiting for a clear statement on climate and climate change, and text that explicitly helps the reader fit 2023 into the broader context of global warming and shifting

fire regimes. This is where this work will end up, regardless of the authors' intentions. The statistics are impressive, and one might think the statistics could "speak for themselves", but the authors have an opportunity to be explicit with how the audience should integrate 2023 in the context of topics like "wildfire crisis," "climate change," "human development", and "forest management." Connect the dots for the readers.

It was not until the last sentence of the paper that a critically important statement is made: "...the 2023 fire season may foreshadow the emergence of severe climate[-]change impacts on fire activity decades earlier than previously anticipated." This seems like a very relevant thesis for the paper itself, but it's not a message that comes through clearly throughout the paper, particularly the "earlier than previously anticipated" point. That has huge implications. Overall, there are several paragraphs where "the lead is buried" a bit, noted below.

Response: We appreciate the reviewer's feedback with respect to potential improvements to the flow and the overall messaging. We made several changes that we feel have addressed this comment, in particular in Paragraphs 2 and 3 of the Introduction (specific changes listed in comments below). We feel like the manuscript is now a lot more forthcoming in terms of linking the 2023 fire season to the anthropogenic climate disruption.

3. The data presented clearly support the statement that "extreme climate enabled the record-setting 2023 fire season." Is that in fact a main point of this work? Full disclosure, I am a fan of the term "climate enabled" (e.g., Higuera and Abatzoglou 2021, GCB). The term clearly attributes climate as a critical contributor, but it does not hang everything on climate, since (as the authors know and communicate well) ignition and extreme fire weather are also needed. All elements are well documented in the manuscript. I encourage the use of "enabled" in several parts of the paper, detailed below.

Response: We accept this suggestion and have added this language, including the expression 'climate enabled' in several places. We were cautious, however, to not be too heavy handed, because a formal climate attribution has so far only been performed for part of the country. See lines 82, 96, 114, 229 and 437.

4. The paper lacks the important context that "fire is a longstanding component of" Canadian forest ecosystems, for millennia. This is simple, but without this, many readers could walk away thinking that 2023 was a crazy climate-change impact entirely. It was not; it was an expression of a longstanding natural phenomenon, under climate change (or "on steroids").

Response: This statement was added to the Material and Methods section, as follows (Lines 453-454): "Wildfires have been a longstanding ecological process in many ecosystems of Canada since the end of the last glaciation."

5. Consider using the term "average" instead of "normal" – simply given the normative use of the term "normal" outside of climatology, statistics, and research in general. This will improve the way this is communicated to non-specialists, particularly journalists.

Response: We agree with this suggestion and made the change in the five places where 'normal' or 'norm' was used (Lines 60, 220, 221, 232 and the Fig. 3 caption).

6. Limit parenthetical statements. If something is important, keep it in the sentence. Limit the use of acronyms, to keep things digestible for a broad audience. Suggestions below.

Response: As suggested, we limited the use of parenthetical statements and acronyms throughout. We have removed the acronym for Ice Mapping System (IMS), persistent positive anomalies (PPAs), and Root Zone Soil Moisture (RZSM), and wildland urban interface (WUI) except where needed for figure appearance.

7. "Data" is used as a singular. Convention is that "data" is plural...but conventions change.

Response: thank you for pointing this out. We have modified this in multiple places (Lines 175, 478, 538, 550, 557, 628, 643, 647)

Detailed comments:

L 54: Consider flipping this sentence to start with "The 2023 wildfire season in Canada was unprecedented in its scale and intensity" – and, to clearly justify "unprecedented" consider simply adding "...scale, intensity, and human impacts". It was 100% unprecedented in human impacts – no arguing about that part.

Response: We have made the suggested change.

L 56: Consider clarifying "periods of dense smoke that caused public health concerns" – e.g., "thousands exposed to hazardous air quality [from wildfire smoke]..."

Response: We modified as suggested, although we used the specific wording "millions exposed to hazardous air quality from wildfire smoke" as the data supports this.

L 58: Take the actual statistic out of the parentheses here – this is a big number that will be cited. E.g., "The record-breaking fire season, with over 15 Mha burned, can be attributed to..."

*Here is the first opportunity to make a clear statement on attribution: both items listed under "can be attributed to" are climate-related: snowmelt and drought. In the authors' view, was the 2023 fire season largely a function of climate? State that; otherwise, readers will wonder about fuels, forest management, human ignitions, etc.

It is true that the record-setting 2023 fire season was...enabled by extreme climate conditions, consistent with expectations of a warming planet? And, perhaps emerging earlier than expected under climate-change scenarios? If so, highlight that in the abstract.

Response: Although we tried as much as possible to accommodate the reviewer's concerns, we were limited by the abstract's strict 150-word limit. We did manage to emphasize the role of climate, albeit with some caution, given that a formal attribution analysis was not undertaken. We made reference to the climate crisis in the following sentence (Lines 58-60): "Anthropogenic climate change enabled sustained extreme fire weather conditions throughout the fire season, as the mean May–October temperature over Canada in 2023 was 2.2°C warmer than the 1991–2020 average."

L 61: Example of replacing "normal" with "average" and removing parenthetical statements: "...was a staggering 2.2 C warmer than the 1991-2020 average, enabling..." And here, these conditions enabled dry fuels – not really extreme fire weather per se, no? When I read "extreme fire weather," I'm thinking about high winds, although I know the FWI integrates fuel moisture. It seems to me that drought (+) created extremely dry fuels, which then enabled successful ignitions and rapid fire spread when winds were high.

Response: We have replaced "normal" with "average" and modified the sentence as suggested.

While it is true that strong winds are often associated with extreme fire weather (specifically in the Fine Fuel Moisture Code, Initial Spread Index, and Fire Weather Index outputs of the Canadian FWI System), extreme fire weather can also refer to extremes in fuel aridity and not only fire behavior indices. For example, during the 2021 heat dome, regions in western North America experienced record high temperatures and low humidity, which led to extremes in vapor pressure deficit (VPD) and the Fine Fuel Moisture Code (See Jain et al. Communications Earth & Environment volume 5, Article number: 202, 2024). Moreover, pyrocumulonimbus events - many that occurred in both 2021 and 2023 - are a function of fuel amount and especially fuel aridity, but tend to occur under lighter winds. In general, the FWI System uses an empirical model to combine several variables (surface temperature, relative humidity, wind speed and precipitation) in a nonlinear way, and extremes in the outputs may occur due to the combination of extreme values in one or more (but not necessarily all) of the inputs.

L 63: Suggestion: "...in the country and across North America" OR "...across North America." As noted, smoke impacts on the NE USA impacted health and prompted much discussion.

Response: We prefer to not generalize to the continent in the abstract, given the focus of the wildfire season in Canada. Also, we are running into the word limit and feel that some of the other concerns raised by the reviewer should take precedence over this suggestion addition. Note that we made sure that the impacts beyond the borders of Canada were properly acknowledged throughout the manuscript.

L 70: "...(c. 1975) over two times the previous record of 6.7 Mha..."

Response: We feel like the current formulation essentially holds the same information and opted to keep it as is. While we made an effort to remove superlatives, we do want to convey the exceptional nature of the 2023 wildfire season.

L 80: The text would be improved with a transition between these first two paragraphs. The second paragraph is highlighting climate as an important contributor to the 2023 fire season, but this is backed into over a number of sentences. The transition from the impacts on communities to climate statistics is sharp. Consider something like: “The 2023 fire season was enabled by record-setting climate. CLIMATE STATISTICS...”

Response: We adopted the reviewer’s suggestion as a topic sentence. Note that an effort was made to improve the transitions among ideas throughout the Introduction. In particular, the link to climate change was strengthened in Paragraphs 2 and 3 to address the reviewer’s main concern.

L 92: “entrained” here seems odd. Was this intended to be “enabled”? That would make sense.

Response: thank you for pointing this out. I believe we were another victim of autocorrect! (it has been fixed).

L 94-95: This seems like the topic sentence (buried at the end of the paragraph), or at least an important topic: increasing wildfire activity in Canada as symptomatic of changing climates. Or, more specifically, symptomatic of global (and regional) warming.

Response: It is true that this general statement could have made an acceptable topic sentence. However, it would have not allowed us to start the paragraph by mentioning the record-breaking climate, which we feel needs to be emphasized early on in the paragraph. In light of this, we rather chose to use the simple, yet impactful, topic sentence suggested by the reviewer (see previous comment).

L 103-105: Again, this seems like a clear topic sentence, hiding in the paragraph: “...2023...was unprecedented and...not anticipated until later in the century based on climate[-]change projections.” This is a really critical point, worth highlighting and leading with.

Response: We respectfully disagree with the reviewer regarding the topic sentence. Although this sentence indeed conveys an important idea, it remains one of the many outcomes that “challenged our understanding”, as stated in the current topic sentences. This said, some unpacking of this broad idea is justified; the text was added to address the reviewer’s concern (Lines 110-111): “However, recent studies report an earlier-than-expected emergence of anthropogenic climate change in many parts of North America (Abatzoglou et al. 2018; Jain et al. 2022).”

L 113: For brevity, consider: “Here we provide an overview...and examine selected drivers and impacts...” Note, only selected driver, and esp. only selected impacts, are presented here.

Consider stating the objective as completed contributions, vs. the “-ing” of each objective. Finally, it’s worth emphasizing the importance of this work in helping contextualize what was an extraordinary event, not only for Canada, but globally. Documenting and contextualizing these impacts of global warming is a critical task more broadly.

Response: We accepted the reviewers suggestions, except that we used ‘main’ instead of ‘selected’ to describe the drivers and impacts. We reframed the latter part of the paragraph to address the reviewer’s concern as follows (Lines 129-132): “We interpret these observations relative to past wildfire seasons, both nationally and regionally. Finally, we discuss the globally relevant impacts of this extraordinary event in the context of rapid climate change.”

L 148: “...since satellite records began in 19XX” – let the reader know when the records began.

Response: We have added “in 1972” to reflect the fact that the first Landsat mission (Multispectral Scanner, MSS) provided data from that year (Line 162).

L 150: The section title could be confusing, with “high” potentially read to refer to altitude. Consider: “Exceptionally large total area burned” or simply “Record-setting annual area burned.”

Response: We have replaced the section heading with “Record-breaking annual area burned”.

L 153: Note that “total burned area” and “area burned” are both used. Either works, but being consistent will help for a general audience.

Response: We agree it would be better to be consistent and have changed two instances of “total burned area” to “total area burned” to be consistent with the majority of cases (see lines 167, 324).

L 154: “...historical national average.”

Response: we have made the suggested change.

L 168: Consider: “...the total area burned (19XX-20XX average – 91%)” – to avoid the reader getting distracted by asking the question “over what period?”.

Response: We added the time period, as suggested (Line 183).

L 182: Is it known, with the statistics on hand, what % of wildfires that resulted in home or structure loss originated from human ignitions? I asked from my own perspective and interests (e.g., Higuera et al. 2023, PNAS Nexus), but I know others would wonder this too. This is critical in the western US, as it clearly points to the fact that most of the “wildfire crisis” (i.e., home and structure loss) is attributable to human ignitions, whereas 88% of wildfires don’t result in any home or structure loss. If not possible with the stats. on hand, no need to dig.

Response: Unfortunately, we do not have these statistics in Canada at this point in time. We used the reviewer's comment to strengthen the argument that human-caused wildfires are responsible for a disproportionate share of the evacuations and structure loss more generally (Lines 190-197): "Human-caused ignitions were responsible for a comparatively low proportion of the total area burned (7%) in 2023, though it is difficult to confidently assign a human cause to some fires due to ongoing investigations and the eventual intermingling of individual fires with multiple ignition sources in large wildfire complexes. These ignitions were numerous early in the season before the greening of vegetation and the prevalence of lightning storms (Parisien et al. 2023) and many became wildfires that burned for months and caused evacuations of communities. Due to their proximity to communities and infrastructure, human-caused ignitions are associated with a disproportionate fraction of evacuations and loss of structures (Tepley et al. 2022; Higuera et al. 2023)."

L 188: Consider "early snowmelt" vs. "premature"? i.e., objectively earlier than average. L 197: "near-average"

Response: We have replaced "premature" with "early" and "near normal" with "near-average" as suggested (Lines 210, 220)

L 209: "hotter and drier...than average"

Response: Modified as suggested (Line 232).

L 210: "...warmer than the 1991-2020 average,..."

Response: Modified as suggested (Line 233)

L 221: Consider: "Fire growth is largely determined by the coincidence of dry fuels, ignitions, and the occurrence of extreme weather conditions..." The text mixes "extreme weather" and "extreme fire weather." I know the latter inherently includes dry fuels, but to the general reader, "extreme weather" is most easily linked to things like high winds.

Response: Modified as suggested. We also added 'fire' to 'extreme weather' in all cases that didn't strictly consider weather, independent of fire occurrence or fire behavior (Line 244).

L 244: Consider spelling out PPA, since the acronym is only used once in this paragraph.

Response: Done (Line 269)

L 249: For brevity, consider keeping figure references in parentheses at the end of the sentence.

Response: We have moved the figure reference to parentheses at the end of the sentence. This is consistent with other figure callouts in the results section. (Line 275)

L 256: Consider: “In 2023, the 834 large fires in Canada was more than 2.5 times the 1987-2022 average of 230.”

Response: We have changed the text to be more in line with this suggestion (Lines 281-282).

L 272-273: Isn't the definition of potential spread days imply “days with $FWI > FWI_{95}$ ”? FWI_{95} was previously defined in the text.

Response: Yes, FWI_{95} was previously defined. For consistency with the earlier definition, we now refer to potential spread days here as “extreme fire weather days” and use the term “potential spread days” in connection with the citation to Wang et al. (2020). The definition in parentheses was replaced with “ $FWI > FWI_{95}$ ” to avoid the redundancy (Lines 298, 299).

L 280: Consider just deleting “bottom-up factors” and just state “may also reflect fuel continuity and topography” as “bottom-up factors” is a fairly fire-specific term.

Response: We replaced the word “bottom-up” with “landscape” (line 307) because we believe there may be other relevant landscape factors other than fuel continuity and topography that are important. In fact, Reviewer #2 has requested an analysis (which we have performed, see below) to determine what types of vegetation burned and if there are any differences that may account for the different observed fire behavior between western and eastern Canada in 2023.

L 294: Subsection title here is a little vague, with “considerations” Consider “response”.

Response: Modified as suggested (Line 320).

L 306-307: Not sure if this holds, but in the US we emphasize that fire/fuel breaks are intended to change fire behavior (including slower rate of spread), but not necessarily halt fire spread.

Response: This is a good point and we have added the words “or slow” to address this (Line 333).

L 314: Strongly suggest deleting “Aside from fire management...” as this qualifier suggests a narrow view of fire impacts. It is well acknowledged that fire impacts – and specifically human impacts – go well beyond fire management personnel. Suggestions: “The human dimensions of wildfire include the vulnerability of communities or individuals to wildfire smoke, evacuations, and loss of homes and structures, and in the worst-case-scenarios human lives.” The “even lives” makes it sounds like a rare possibility – it’s happening more and more each year. In fact, it is remarkable – literally worth noting in the text – that the death toll from the 2023 fire season in Canada (not in the US) was kept so low.

Response: We have removed the qualifier “Aside from fire management” and accepted the reviewer’s suggested change (Line 339). In the modern era, the death toll from wildfires has been relatively low in Canada when compared with other countries with active fire seasons; this is due to a combination of factors, not least being the relatively low population density in Canada, as well as the management of risk through the early and proactive issuance of evacuation orders. While we agree with the reviewer that the low death toll is noteworthy, we prefer to not mention it because it is an extremely sensitive topic in Canada and may be perceived by some as diminishing the gravity of the loss or lives that did occur.

Consider citing, in the first sentence of this paragraph, a paper on human dimensions of wildfires and vulnerability: “Social drivers of vulnerability to wildfire disasters: A review of the literature: <https://www.sciencedirect.com/science/article/pii/S0169204623001160>

Response: As suggested, we have added this citation to the end of the first sentence of that paragraph (Line 341).

L 332: Over 3x more?

Response: This is true, but in this case we do not believe it is necessary to further qualify the statement.

L 381: For the ecological impacts of the 2023 wildfire season, consider citing Coop et al. (2020), for a broad, recent, North American overview:
<https://academic.oup.com/bioscience/article/70/8/659/5859066>

Response: This reference was added to the last sentence of the paragraph (Line 420).

L 387-391: This is all true, but generally a weak way to link the 2023 fire season to climate change, global warming. Aren’t we past the stage where we wonder if we’re seeing the impacts of global warming manifest through wildfire activity? Consider something like “Although future research is necessary to quantify the degree to which anthropogenic climate change facilitated the 2023 fire season, it is clear that extreme climate conditions, all consistent with expectations of global warming,...” This is my opinion, but I think it’s a disservice if this text comes off as wondering if climate change was a factor here; details and specifics, yes, but overall, this is consistent with what we’ve expected and predicted, since Flannigan and Van Wagner 1991 (!).

Response: With a step back, we fully agree with the reviewer. We removed the somewhat tepid text underlined by the reviewer and replaced it with the following sentence (Lines 428-430): “Anthropogenic climate change facilitated the 2023 fire season, which is consistent with expectations of global warming put forward over the past several decades (e.g., Flannigan and Van Wagner 1991).” In addition, we strengthened the text related to the effect of global warming and included it higher up in the manuscript (see earlier comment).

L 396: "...moisture deficits enabled the record-breaking 2023 fire season." This fits under the framework of "climate enabled, weather driven" as an explanation for extensive fire activity.

Response: We replaced 'drove' by 'enabled', as suggested (Line 437).

L 402: This suggestion of prioritizing fire prevention is just doubling down of fire suppression? This seems odd, not to mention very dangerous.

Response: We replaced 'prevention and mitigation' by 'risk mitigation'. In this context, prevention refers to the avoidance of human-caused ignition, which, as pointed out in the text, causes significant problems every year in Canada (as in most fire-prone areas). However, given that the term can mean different things to different people, we opted to remove it.

L 404-405: As note above...this is a HUGE and very important point. The paper would benefit if this was highlighted throughout, and expended upon a bit more.

Response: This comment is related to an earlier comment. We made an effort to emphasize this point throughout the manuscript. To strengthen the statement, we modified the end of sentence as follows (Lines 445-456): "[...] as the 2023 fire heralds the emergence of severe climate change impacts on fire activity decades earlier than previously anticipated."

L 409: "Wildfire in Canada" – this is at least one place where it should be noted that wildfire in Canada is a longstanding process, for millennia.

Response: We have added the following sentence (Lines 453-454): "Wildfires have been a longstanding ecological process in many ecosystems of Canada since the end of the last glaciation."

L 485: This page of method has a lot of acronyms. If the acronym is only used once or twice in a paragraph, consider eliminating it and just spelling out the term, for clarity.

Response: We have removed acronyms that were used only once or twice, as suggested.

L 519: I'm familiar with the CFFDRS...but I don't know what "screen-level temperature" are.

Response: Screen-level meteorological measurements are usually performed at 1.3-2 m in the presence of a shelter (eg. Stevenson screen) to shield measuring apparatus from local sources of precipitation or heat. However, the point is taken that perhaps this is unnecessary detail, particularly because this term is not relevant when using reanalysis data to compute fire weather indices. We have therefore replaced the words "screen-level" with "surface" (Line 570).

L 534: What size is the "grid-level" here?

Response: As mentioned earlier in the text, each grid cell is at 0.25 degree resolution for the ERA5 reanalysis data, which corresponds to approximately 31 km at the equator. To extract the exact physical size of each grid cell for a specific location would require a coordinate transformation with a latitudinal adjustment to account for the fact that grid cells are smaller for increasing latitude. Here, the correct transformation with area adjustment was used when computing the extent of extreme fire weather. However, we do not believe it is necessary to add this rather technical detail here as this is a standard procedure that is performed with built-in tools available in most GIS software. In fact, the words “grid-level” are not required here so we have now omitted them.

Figures

Figure 1: Strongly suggest changing the text (in the upper-right corner) to “...largest extent of land impacted by fire in Canada’s history” less you perpetuate the misconception that wildfires destroy all land they burn across. The land is not “consumed” – it’s still there (thankfully)!

Response: Modified as suggested.

*For Fig. 1 and 2 – are these statistics for “total area” or just “forested area”? I think the former...but specifying would be helpful.

Response: This refers to the total area, which, in Canada is quite close to the forested area (See the new Supplemental Table S2). Fig. 1 has the text “Total area burned by province” to make this clear.

*Fig. 3: Suggest using “average” here too instead of “norm”.

Response: Changed as suggested.

*Fig. 4: Including titles after the panel labels (“a),” etc.) would be helpful here, as in Fig. 3.

Response: Added as requested. We have also slightly modified the figure caption to clarify that the rankings are for 2023 values relative to the period 1940-2023.

*Fig. 5: Great figure! Is it possible to have a y-axis for each panel, at least for one of the subpanels? There’s no ways gauge the value of area burned (a) or number of days (b) currently.

Response: We have added a second axis to both panels a and b as requested. We have only added this axis to the top (Quebec) curve as to not crowd the plot although the scaling is the same for other curves except where the multiplicative factor has been applied (as indicated). Note that panel b represents the forested area in which FWI is exceeding the 95th percentile of FWI, and not the number of days.

Fig. 6: Having panel (a) as an anomaly (relative to average) as well would be useful. It would arguably be more useful than panel (c) in this figure (which could go to the appendix). That would keep this as a four-panel figures.

Response: We have replaced panel (c) with the suggested figure (now panel b) and modified the caption accordingly. The previous blocking figure has been moved to the supplementary as a new figure (S2) with subsequent supplementary figure numbers incremented and figure callouts appropriately changed. We also added the text to methods to clarify that days with positive persistent anomalies (in 500-hPa geopotential heights) are being used as a proxy for blocking days (blocking occurring when large-scale pressure patterns are nearly stationary for several days) . On revisiting the data, we also decided to add the mean number of blocking days across Canada, which was an extraordinary ~50 compared with the mean value ~15; this was explicitly stated (see Line 274).

Fig. 7: In my opinion, this entire figure could go to an appendix.

Response: We respectfully disagree with the reviewer. One of the main features of the 2023 fire season was the large size of fires—many wildfires were immense (>500,000 ha). As a result, the 2023 fire size distribution diverted wildly from the average distribution. One of the reasons for so many large wildfires is that many of them started early in the season and burned for several months, a phenomenon that is not often described in the literature and it, in our opinion, was worth highlighting.

Fig. 8: Consider having panel (b) as an anomaly, instead of the PM2.5 values. That would help highlight what's unusual about 2023.

Response: Although this would help emphasize how exceptional 2023 was, we feel that the absolute value of PM2.5 is a relevant metric to display. Extreme values of PM2.5 correspond with elevated health impacts, and an anomaly map would unfortunately downplay the impacts felt by northern communities, which are more regularly exposed to smoke events.

It is worth noting that since PM2.5 is a known pollutant, countries (including WMO) have established absolute concentration/exposure limits:

- Canadian's CAAQS is 27 ug/m³ (24hr avg) <https://ccme.ca/en/air-quality-report#slide-7>
- USA's NAAQS is 35 ug/m³ (24hr avg) <https://www.epa.gov/criteria-air-pollutants/naaqs-table>
- WMO is 15 ug/m³ (24-hr short term AQG level) <https://iris.who.int/handle/10665/345329>

Analysis of the PM2.5 anomaly is nonetheless a good suggestion and we hope to consider it in a future study.

Reviewer #2 (Remarks to the Author):

Comments for the Manuscript NCOMMS-24-16816 “Canada Under Fire – Drivers and Impacts of the Record-Breaking 2023 1 Wildfire Season”

Summary:

The authors present an overall assessment of the record breaking 2023 fire season for Canada, providing information about the causes, the impacts, and the implications. Throughout the manuscript, authors provide quantitative information to support their analysis, based on several lines of evidence, datasets and literature. Authors provide a comprehensive overview of the wildfire season, with regional statistics referring to the burned area and the drivers that led to it. They effectively summarize the drought and fire weather conditions that led to the exceptional fire year, and present considerations referring to the fire management intensity and its implications to the burned area. Finally they consider socioeconomic impacts of the fire season. The manuscript is well written and well organized, while the figures (except for some minor comments provided here) are of great quality.

Response: We are grateful for these positive comments.

Main comments:

While there is not much to criticize in this work on what is already in the manuscript, what seems to be missing is a good insight into the vegetation that was burned, i.e. types of trees/forest per region, as well as the vegetation conditions prior the wildfire events compared to the climatology. Information about the types of vegetation may also partly explain the regional differences between the Eastern and Western Canada. Regarding the vegetation conditions, while the DC code of the FWI partly covers this aspect, the DC is a very simple soil moisture accounting scheme and hence can provide an estimate on the “agricultural” drought status, but not the vegetation – despite the fact that the FWI was initially created for Canadian wildfires. A suggestion would be to expand their analysis into a remotely sensed index (like NDVI if there is not any other better?) as a comparison between the climatological values and the 2023 values.

Response: We agree that more information pertaining to the vegetation type that burned and a more in-depth characterization of vegetation condition prior to burning would add a useful dimension to this overview of the 2023 fire season. Although we added an analysis to partially address the reviewer’s concern, it was beyond the scope of this study to undertake a detailed analysis of the fine-scaled factors that drive fire spread.

We computed the vegetation type that was burned by provinces and territories to make a simple comparison among areas (see new Table S2). The data shows that the vast majority of the area burned occurred in forest systems (87.5%). A larger proportion of non-forests burned in areas where other fuel types are common or dominant, such as in the tundra (eg, Nunavut) or peatland-rich areas (eg, Newfoundland and Labrador); however, in relative terms there was little fire activity in these areas. We added the following text to support these new results (Lines 397-401): “Nationally, wildfires predominantly burned forests (87.5%), except in ecotonal areas that transitioned to grassland ecosystems in the south (i.e., herbs vegetation type) and those that transitioned to tundra in the north (i.e., herbs and shrub types) (Table S2). Wildfires chiefly

burned conifer and mixedwood forests, but a substantial area of broadleaf forests burned in Alberta due to this vegetation type regional dominance and the propensity of springtime, pre-greenup burning (Parisien et al. 2023b).”.

As for the vegetation conditions, what the reviewer suggests represents a massive undertaking that would be best addressed in one or more follow-up studies. Although the data exists to examine the daily condition before or during a given wildfire, doing a proper analysis would require an examination of all the main factors that control wildfire ignition and spread (including topography, land cover, and the spatial configuration of these factors on the landscape). This said, some important general inferences can be made from the analysis of drought and prior weather conditions (e.g., Fig. 4 and 6) and the timing and seasonality of ignition (Fig. 7).

Similarly, the suggested NDVI analysis is not without challenges. In many northern landscapes, NDVI varies substantially among vegetation types; therefore, the correspondence between on-the-ground vegetation and the spectral indices must be established. For instance, compared to forests, a rich wetland (e.g., an open fen) can have a very high NDVI without having a particularly impressive biomass load. An even greater challenge is depicting the intra-annual variability observed in these landscapes. In most Canadian landscapes, there are strong phenological signals, but also important fluctuations related to drying/wetting in wetlands. This is not to say that these indices are not useful, but the work that would be required to meaningfully interpret them in different fuel types, seasons, and topo-edaphic settings is beyond the scope of this study.

Subchapter “Fire size and duration”: Authors define the potential spread days as the days where the FWI exceeds the 95th percentile of May-October climatology. But is this true? To what I can interpret from Fig S6, in many cases the fire sustains (while not flourishing) for long periods (in days) with pyro-meteorological conditions below 95th percentile. So the question that arises is whether there is a rough threshold in the FWI that can be considered as a “fire-ending event”?

Response: This apparent discrepancy can be understood in the context of “spread days”; extreme fire weather days (whether defined by FWI exceeding the 95th percentile or some other threshold, see for example Wang et al. 2023) are related to daily linear fire spread and not total daily area burned. As a fire grows large, relatively small linear growth can lead to large area burned due simply to geometric considerations (specifically, wildfire growth through time follows a power function).

In terms of a FWI threshold for a fire-ending event, this needs to be considered probabilistic, because no single set of conditions can be confidently considered ‘fire ending’. For example, 20 mm or rain could easily decrease the potential for ignition or spread of a wildfire, but may not be enough to extinguish a fire smoldering in deep organic soils (i.e., peatlands) that may rekindle (eg. holdover or overwintering fires). This said, a combination of factors may be preferable in defining a “fire-ending event”; however, this is outside the scope of the present work, which was to provide an overview of the drivers and impacts of the 2023 fire season.

Other comments

L224: Figure 5 does not seem to provide any correlation information between FWI and BA. There is obviously a covariation in both quantities. It would be interesting to also provide some type of correlation (Pearson or Spearman maybe?) . As also stated in the manuscript, Fire weather index can correlate well with the logarithm of the BA.

Response: We have calculated the spearman correlation coefficient, which being a rank based quantity gives the same correlation coefficient between area $FWI > FWI_{95}$ and BA or the logarithm of BA. We have added this as panel c of Fig. 5 and made reference to the fact that the correlations are larger in western Canada than in the central and eastern parts of the country in the main text (Line 253).

In figure 3a, the snowmelt timing pattern in the central Quebec region, shows a rapid transition between May and ~Jun values. Is there any elevation/aspect/soil/climate reason for that? The same reason also causes the same pattern in Figure 3b, and for some reason partly in the Figure 3f (June DC) but interestingly not in the Figure 3e (May DC)? Authors should check for potential issues in the data or their processing.

Response: The data has been double checked, and in so far as the data source is accurate, there are no problems with the data. Although it appears as a discontinuity, it should be noted that the reason for this is simply that this corresponds to the pattern of snow on the ground in late May (see for example, the snow map for May 24 below). It is also not surprising that this pattern is therefore evident in the May DC as this is the month in which the snow melted for this region (the presence of snow would affect several inputs to the DC calculation including temperature and overwintering adjustment to precipitation). The discontinuity disappeared by June after the snow had fully melted. Indeed, the fact that this feature is shown in two independent datasets lends further confidence to the results shown.

Figure 4: Authors should add titles in the sub-figures of Figure 4 as they have done in Figure 3. Also Figure 4 caption does not make any sense, something is going wrong.

Response: We have added caption titles as per Fig. 3. Similar captions have also been added to Fig. 6 for consistency (but not Fig. 7 which does not display maps). Also, thank you for alerting us to the problem with the caption for Fig. 4; there was an issue with the references to the individual panel figures that is now fixed.

Figure 5: What is the x_2 and x_{10} values mentioned at the right of Figure 5a? This should be mentioned in the figure caption. Also, is there any way to include y-axis information?

Response: We explained the x_2 and x_{10} in the caption. As also suggested by Reviewer #1, we have added a second axis to both panels a and b as requested. We have only added this axis to the top (Quebec) curve as to not crowd the plot although the scaling is the same for other curves except where the multiplicative factor has been applied (as indicated).

REVIEWERS' COMMENTS

Reviewer #2 (Remarks to the Author):

Comments for the Revised Manuscript R1-NCOMMS-24-16816 “Canada Under Fire – Drivers and Impacts of the Record-Breaking 2023 1 Wildfire Season”

Dear Authors

Thank you for your response and the additional analysis provided. I appreciate the effort made to address my comments, especially the inclusion of the Table S2 and the supplementary text which enhance the manuscript's insights into the vegetation types burned. I understand and acknowledge the constraints mentioned regarding a more detailed analysis of vegetation conditions prior to burning and the challenges associated with implementing a remotely sensed index like NDVI. While I still believe that the amount of additional work doesn't justify one or more follow-up studies, I am content with the revisions and the overall response. Therefore, I agree to proceed with the publication as it stands. Thank you for your attention to my feedback and the diligent work in revising the manuscript.

All the best